# *ADABase*: A Multimodal Dataset for Cognitive Load Estimation

**DOI:** 10.3390/s23010340

**Published:** 2022-12-28

**Authors:** Maximilian P. Oppelt, Andreas Foltyn, Jessica Deuschel, Nadine R. Lang, Nina Holzer, Bjoern M. Eskofier, Seung Hee Yang

**Affiliations:** 1Department Digital Health Systems, Fraunhofer IIS, Fraunhofer Institute for Integrated Circuits IIS, 91058 Erlangen, Germany; 2Machine Learning and Data Analytics Lab (MaD Lab), Department Artificial Intelligence in Biomedical Engineering, Friedrich-Alexander-University Erlangen Nuremberg, 91052 Erlangen, Germany; 3Department Sensory Perception and Analytics, Fraunhofer IIS, Fraunhofer Institute for Integrated Circuits IIS, 91058 Erlangen, Germany; 4Artificial Intelligence in Biomedical Speech Processing Lab, Department Artificial Intelligence in Biomedical Engineering, Friedrich-Alexander-University Erlangen Nuremberg, 91052 Erlangen, Germany

**Keywords:** cognitive load, affective computing, autonomous driving, machine learning, multimodal dataset

## Abstract

Driver monitoring systems play an important role in lower to mid-level autonomous vehicles. Our work focuses on the detection of cognitive load as a component of driver-state estimation to improve traffic safety. By inducing single and dual-task workloads of increasing intensity on 51 subjects, while continuously measuring signals from multiple modalities, based on *physiological* measurements such as ECG, EDA, EMG, PPG, respiration rate, skin temperature and eye tracker data, as well as *behavioral* measurements such as action units extracted from facial videos, *performance* metrics like reaction time and *subjective* feedback using questionnaires, we create *ADABase* (**A**utonomous **D**riving Cognitive Load **A**ssessment Data**base**) As a reference method to induce cognitive load onto subjects, we use the well-established *n*-back test, in addition to our novel simulator-based *k*-drive test, motivated by real-world semi-autonomously vehicles. We extract expert features of all measurements and find significant changes in multiple modalities. Ultimately we train and evaluate machine learning algorithms using single and multimodal inputs to distinguish cognitive load levels. We carefully evaluate model behavior and study feature importance. In summary, we introduce a novel cognitive load test, create a cognitive load database, validate changes using statistical tests, introduce novel classification and regression tasks for machine learning and train and evaluate machine learning models.

## 1. Introduction

The rapid development of novel sensor technology, powerful computing capabilities and methods using artificial intelligence has moved the prospect of autonomously driving vehicles into a potential candidate to transform the way people experience mobility. This development is a promising direction for traffic safety. Surveys [1,2,3] reporting the causes of traffic crashes in manual driving, have identified cognitive and emotional load among others as a major factor. However, until fully autonomous driving is available, the operator still needs to observe and, in critical situations, take control of the vehicle. To measure the required degree of manual interaction with the vehicle, the Society of Automotive Engineers (SAE) defines 6 levels of automation ranging from 0 (fully manual) to 5 (fully autonomous) [4]. While even lower level autonomous driving can reduce the complexity of traffic situations and therefore lead to task execution at levels with lower cognitive load [5], recent studies suggest, that inattention or distraction through additional tasks performed by the driver can lead to accidents for vehicles of level 1 (assisted driving), level 2 (partial driving automation) and level 3 (conditional driving automation) [6,7]. When performing a secondary task, for example, talking to a car passenger while driving, inattention to the primary task, in this example, driving a car, can manifest itself on different levels: visual (not monitoring the road or the vehicle), manual (hands not on the steering wheel) or cognitive components [6], while visual and manual inattentions are important components for driver monitoring, we study the detection of phases with high cognitive states.

These states are of special interest, since engaging in dual- or multitask settings, a driver’s attention is a finite resource and cognitive limitations are soon met [8]. The research community describes a subject’s, cognitive state, with different terms like cognitive workload, mental workload, cognitive load or task load. We use Working Memory (WM), which is based on Baddeley’s model [9], as storage of conscious information that is limited by the amount of information one can hold and process. Following this definition, we use Cognitive Load (CL) as a measurement to define the amount a subject puts on the WM during a task [10]. The definition of CL by [11] as a “… multidimensional construct representing the load that performing a particular task imposes…” indicates that the prediction of when a person can attend to information (or process additional workload) is challenging.

One can study the concept of CL empirically, through meaningful measurements from four categories [12,13]. *Subjective measurements* can capture Perceived Mental Workload (PMWL) as shown in early studies, that used a mental-effort rating scale with 9-grades as symmetrical categories from very, very low mental effort (1) to very, very high mental effort (9) [14], while recent studies often use a multidimensional measure to evaluate perceived CL such as NASA-Task Load Index (TLX) [15,16,17]. Even though researchers have put a lot of work into developing these Likert-scale-styled subjective rating systems for different tasks, major issues remain: (a) subjective ratings rely on the participants’ ability to introspect their current cognitive load [18], (b) the current task needs to be interrupted to answer the questionnaire and (c) subjective feedback can not be measured in real-time and continuously. *Performance measurements* are strongly related to the task imposed on the subject. Several metrics have been used in the past, such as the number of mistakes made during a complex learning task [19] or more general accuracy metrics such as hit-rate or reaction time, when solving standardized tasks such as *n*-back [20]. Others have used a dual-task load paradigm and evaluated the performance on the secondary task as an indicator of the cognitive load induced by the primary task [20]. In the area of driver monitoring, Engström et al. [6] reviewed the literature and identified inconsistent and counterintuitive findings, with an increasing, as well as a decreasing number of accidents, as a performance metric under increased cognitive load imposed by secondary tasks such as speaking while driving or listing to the radio. The measurement of CL using performance metrics is therefore limited by being strongly selective and task-dependent. A major advantage of performance metrics, such as response time of braking on brake light onset of leading cars, is, that these can be continuously measured during real-world driving situations [6]. *Behavioral Measurements* such as linguistic [21], speech [22,23,24], device inputs like pen [25,26] or computer-mouse movements [27] are commonly used in various applications, while metrics computed from behaviors such as lane keeping, steering wheel activity, mean vehicle speed or the number of initiated speed changes are more specific to driving [6]. A very prominent approach in the affective computing community is the extraction of action units from facial videos to detect emotions. These action units can also be useful to detect cognitive load while driving [28,29]. Another extensively researched modality is based on eye-tracking technology [30,31], that in addition to the behavioral dimension accounts to physiological measurements’ with parameters such as pupil diameter. *Physiological Measurements*, include other properties of the eye, such as pupil diameter or pupil change responses [32]. Other physiological processes of the heart, muscles, lung, skin (temperature or conductance) and brain can be measured using biosignal acquisitions [33]. Both physiological and behavioral measurements have therefore the major advantage of being able to be recorded continuously. In addition to these biosignal recordings, biomarkers such as amylase or cortisol, commonly measured in stress studies, may help to detect phases of high cognitive load [34].

Our attempt to measure cognitive load using multimodal data is influenced by other areas of research, that try to detect psychological states, especially contributions in the area of affective computing. Lisetti et al. have summarized early work in a review article, highlighting several contributions that show statistically significant changes in different elicited emotions [35]. Wang et al. attempted to recognize five emotional states of various drivers in an automotive simulator application setting [36]. With recent advances in machine learning over the last years, not only novel algorithms have been developed, but the requirement for high-quality data and therefore the collection and annotation process gained attention. Datasets such as DEAP [37], DECAF [38], cStress [39], ASCERTAIN [40], uulmMAC [41], AMIGOS [42] and WESAD [43] have been published in the affective sensing community.

In addition to the study of emotions and stress, other researchers already studied the concept of cognitive load while driving using multidimensional measurements. Haapalainen et al. studied the effect of various elementary cognitive tasks while recording the psycho-physiological activity using wearables [44]. Hussain et al. compared different modalities (videos of the face and electro-physiological measurements) for different cognitive load tasks and proposed a fusion technique of inputs [45]. Closely related to our contribution is the distracted driver dataset published by Taamneh et al., that records various driving conditions in a simulated environment [46]. CLAS [47] and MAUS [48] published datasets enabling the study of multidimensional data during different simulated cognitive load tasks. A noteworthy contribution, that shares a close relationship with our work on inducing cognitive load to drivers, is the eDREAM dataset [49,50]. A more practical approach for avoiding crashes through automatic measures is presented by Healey and Picard [51], that analyze cognitive load during real-world driving tasks using physiological signals. Another real-world driving study, conducted by Friedman et al. [52], tried to estimate cognitive load using only non-contact sensors during two experiments, a driving experiment and a version of an *n*-back task based on datasets, that have been introduced in [53,54].

Aside from the research gaps, described below and answered throughout this study, the analysis of related datasets makes it apparent, that issues with current data acquisitions setups and datasets remain:Some datasets are not publicly available to the research community. We release recordings of 30 subjects.Related work collected combinations of input modalities from subsets of relevant measures, while our work combines a diverse set of modalities form *physiological* and *behavioral* measurements, *subjective* questionnaires and *performance* metrics.In recent years, researchers introduced and studied various tasks, however, we as a research community failed to provide a database where the same subject participated in different tasks with different intensities of cognitive load in one recording session. Our setup fills this gap and therefore enables the analysis of distribution shifts and the evaluation of the robustness of predictors with representations that generalize across tasks and test the effects of subject-wise shifts during analysis.Missing information about metadata of the sample population like age, sex and personality traits, as well as the potential inclusion of subjects with undetected medical conditions or medication treatments, that might interfere with some measurements.

In this work, we focus on the estimation of cognitive load, collecting measurements of all four groups: physiological measurements, performance measurements, subjective measures and behavioral measures. For our subjective measurements, we employ the multidimensional NASA-TLX test. For performance measurements, we measure based on correct and incorrect hits recall and precision for our tests, as well as the reaction time. Our physiological measurements include Electrocardiography (ECG), Electrodermal Activity (EDA), Electromyography (EMG) Photoplethysmogram (PPG), respiration and eye tracker data. For behavioral measurements, we use action units extracted from video recordings. We enrich these data records through a detailed analysis of our study cohort design. To explore the potential use of cortisol as a biomarker for CL, prominently used in stress studies, we collect salivary samples. Furthermore, our statistical analysis of various extracted statistical and medical-motivated expert features provides explainable measures of cognitive load. We train various machine learning models to predict CL of subjects while participating and observing a close-to real-world driving simulation of an autonomously driving car. This leads to our key major contributions:We implemented a simulation environment for autonomous driving software to induce different levels of cognitive load and a fully synchronous network of recording devices for multimodal measurements.We create a dataset that provides a wide range of physiological modalities, subjective ratings, performance metrics, behavioral data and biomarkers with precisely annotated phases of multiple levels of cognitive load.We conduct a robust statistical evaluation, and present various statistical and expert features with significant changes for multiple modalities.We identify several meaningful combinations of modalities to measure cognitive load using multimodal fusion techniques.We propose a novel continuous cognitive load value, combining subjective and performance measurements as a target for training.We release **A**utonomous **D**riving Cognitive Load **A**ssessment Data**base** (*ADABase*), containing 30 subjects to the public to enable the development of novel algorithms for multimodal machine learning.

In addition, research gaps in related works are clarified by providing a detailed analysis of 51 subjects with multimodal data recordings including modalities that are only briefly studied, such as action units and cortisol measurements at different levels of cognitive load. Furthermore, we combine multiple modalities that are until now, studied in subsets in related work, to a superset of modalities for cognitive load estimations, including remote measures such as eye tracking data and facial videos. We have created a new simulation with a close-to-real-world autonomous driving scenario, while also recording the extensively studied *n*-back test for easier alignment with related work.

## 2. Materials and Methods

To study the concept of cognitive load and develop machine learning algorithms, that utilize multimodal data for the detection of different cognitive load levels, this study focuses on the implementation of a fully integrated driving simulator. Our simulator includes the ability to record a wide range of physiological signals, face videos and eye tracker data, performance data, task-specific subjective feedback self-evaluations, personality traits and stress-related questionnaires while ensuring that all data points are synchronized across multiple acquisition platforms and modalities. In this study, we induce CL in two different ways. The first test is motivated by recent advances in the development of autonomously driving vehicles. We developed a test with subject/test-system interactions, that are similar to driver/vehicle interactions in lower-to-mid-level-autonomous vehicles. This test is introduced in Section 2.3 and contains different levels of cognitive load and the addition of a secondary task in the form of controlling an entertainment system while observing the vehicle. The second test is well-established in the research community: The *n*-back test introduced in Section 2.4 is conducted at three levels of difficulty and with single (visual) and dual-task workloads. However, due to the basic concept of our scenario and the conducted psychological assessment (*n*-back), commonly used in cognitive neuroscience to measure the working memory and its capacity, the results may generalize well to similar scenarios.

One of the many development contributions in this dataset is the usage of high-precision multimodal recordings and coordinated cognitive load simulation. The simulator for both tests is introduced in Section 2.2. In Section 2.1, we introduce statistics about our cohort design and relevant parameters such as driver experience. Our approach to measure the response of the hypothalamic-pituitary-adrenal (HPA) axis using salivary cortisol values is introduced in Section 2.5. Methods to evaluate subjective feedback are introduced in Section 2.6 and performance measures and metrics are described in Section 2.7. Section 2.8 describes our physiological measurements equipment and recording setup, including relevant references to current literature and a list of extracted expert and statistical features. The same information is provided for eye tracking data in Section 2.9 and behavioral data extracted from videos in Section 2.10. The handling of artifacts is described in Section 2.11. Before using the extracted expert features, motivated by medical practitioners and affective sensing literature, for the detection of phases with different levels of cognitive load we introduce our statistical evaluation protocol in Section 2.12. In Section 2.13, we introduce machine learning tasks using different sets of modalities for simple binary classification between low and high levels of cognitive load. Additionally, to this simple binary classification, we propose a three-class under-to-overload classification task and use subjective feedback and performance metrics to develop machine learning algorithms with a continuous cognitive load level as output. Our machine learning pipeline and algorithms are introduced in Section 2.14.

### 2.1. Participants and Cohort Description

The demographic data of all participants was acquired through self-reporting. Reported parameters were age, sex, weight, height, the highest obtained educational degree, state of employment, first language, state of driving license, as well as the duration of driving experience and handedness. Subjects with medical conditions or subjects under medications, that are known to have an impact on behavior, cognition, physiology or the HPA axis analyzed in this experiment, have been excluded upfront.

The acquired dataset consists of 51 (24 female, 26 male subject and 1 subject that did not want to state gender and age) subjects with an average age of 26.53 ± 5.93 years, where the youngest subject is 18 years and the oldest subject is 42 years old. The distribution of male and female participants is visualized in Figure 1. We measured body weights between 50.0 and 108.0 kilogram, with a mean weight of 71.2 ± 13.5 kg and a body height between 1.58 and 1.93 meters with a mean of 1.75 ± 0.09 m. We computed the Body-Mass-Index (BMI) yielding 22.9 ± 2.8 kg/m${}^{2}$ in average, with a minimal value of 18.4 and a maximal value of 30.9 kg/m${}^{2}$. Following the World Health Organization (WHO) expert consultation classification recommendations for adults, described in [55], we identify 1 subject as underweight (BMI <18.5), 37 are in normal range (18.5≤ BMI <25), 12 subjects are pre-obese (25≤ BMI <30) and one participant was obese (BMI ≥30). We publish this meta information with our dataset to enable future work to develop algorithms that take the subject meta information into account. Eight of 24 female subjects were on contraceptive medications. Two subjects were using Levothyroxin to treat hypothyroidism. As our study involves various driving scenarios, we asked the number of years, since the driving license was acquired. Subjects without driving licenses were set to zero years of experience, yielding an average driving experience of 8 years of all participants. The detailed distribution is shown in Figure 1. Of 51 participants, 49 were right-handed and two left-handed. The experimentation setting was adjusted to match the handedness. The subjects were recruited from a diverse population of employees at the Fraunhofer Institute for Integrated Circuits and psychology students at Friedrich Alexander University Erlangen-Nuremberg. One complete session in our simulator, including preparation, measurements and debriefing took, depending on the subject’s compliance, around three hours, with an average signal recording time of 155 ± 13 min per subject. The data was acquired on workdays from Monday to Friday at different times of the day. All subjects gave their informed consent for inclusion before they participated in the experiment. The study was approved by the Ethics Committee of the Friedrich-Alexander-University Erlangen Nuremberg. (Ethics-Code: 129_21 B) The complete experimentation recording was 135 h for 51 subjects. We acquire psychological profiles as suggested by Gjoreski et al. when evaluating mental workload through self-reported questionnaires [56]. To analyze the impact of personality, we utilize the Big Five Inventory (BFI) as a model for the description of personality. To reduce testing time we use a short version, called Big Five Inventory-Kurz (BFI-K) introduced in [57] as an assessment questionnaire. We report the results for all five traits: extraversion vs. introversion, agreeableness vs. antagonism, conscientiousness vs. lack of direction, neuroticism vs. emotional stability and openness vs. closeness.

### 2.2. Experiment Structure and Simulator Environment

The experimentation schedule consists of two main test phases, separated by a questionnaire phase to acquire self-reported subjective measures, targeting mental-well being, chronic stress and all demographic information reported in Section 2.1. The study setup was conducted in a controlled environment, to ensure, that no interruptions occur. An approximately constant room temperature was held during the completion of the experiments. To mitigate the effects of different body positions on parameters like Heart Rate Variability (HRV), the subject was in a sitting position in a car seat. The light conditions were the same for all subjects and no windows were present in the room (no sunlight). Issues with the circadian rhythm of the test subjects are taken into account, as all tests were conducted in fixed time slots, starting either at approximately 9 a.m. or 1 p.m. During this study we conducted two main tasks, in a randomized order:*n*-back Test: A established and commonly used continuous assessment of the wm, described in Section 2.4 [58].*k*-drive Test: An application task with multiple levels of cognitive load with a primary and secondary task, while observing an autonomously driving vehicle, described in Section 2.3.
During the complete experiment, the subjects were located in the center, sitting in an upright position in a simulator car seat (Playseat ®Evolution Pro Gran Turismo, Playseat B.V. Innovatieweg 18-19, 7007 CD Doetinchem, The Netherlands). The user inputs were entered using the button controls of a steering wheel (Logitech G29 Driving Force, 7700 Gateway Blvd. Newark, CA 94560, USA). The different tasks were visualized using a multi-monitor setup (four 4K 55inch monitors) to enable an immersive experience and a simulation environment that is motivated by real-world car cockpits. All commands, instructions and custom tests were shown on the top monitor, while the three monitors below were used to show the driving simulation. The tablet was located on the side of the dominant hand. The test system was running depending on the active study phase, several custom applications, which were written in PsychoPy [59], exectued on a separate test system that is synchronized to the Biopac (BIOPAC Systems Inc., 42 Aero Camino, Goleta, CA 93117, USA) system, described in Section 2.8 using a Universal Serial Bus (USB) to TTL connection. A camera system, described in Section 2.10 and an eye tracking device, described in Section 2.9 were located above the steering wheel. The camera system is synchronized using analog triggers from the Biopac system and the eye tracker was connected to and synchronized with the PsychoPy test system. Images documenting the test environment are shown in Figure 2. An overview of the technical setup is shown in Figure 3.

The experimentation software is developed using PsychoPy version 2021.1.0. All experiments and user instructions are described in German language within the software. To mitigate any effects that can arise from personal interactions between the examiner and the subject only a few interactions are required during the experimentation phase. At the beginning/end of the experiment, the subject is connected/disconnected to the Biopac system. During the questionnaire phase or before the driving test starts, the tablet is positioned by the examiner to ensure ergonomic accessibility. In addition to these interactions, the salivary samples are collected given the described schedule in Section 2.5. Deviations from the schedule are documented by the examiner on a predefined template.

### 2.3. Autonomous Driving Simulation

We are simulating a semi-autonomous driving experiment, where only little interaction of the subject with the vehicle is required and increasing levels of single- and dual-task load is put onto the subject. The steering, gas and brake control as well as gear shifting is autonomously controlled by the simulator. The subject’s interaction is limited to three buttons, located on the steering wheel to detect three different maneuvers as a primary task. During level 1, the subject can use the 
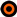
 button to indicate that the autonomously driving car is passing another car (overtaking). In level 2, the subject is additionally asked to indicate that the car is being overtaken, which can be detected by pushing the 
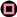
 button. For the last level (3), events of high acceleration and deceleration need to be detected by pushing the 
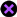
 button. All driving sessions were simulated using the playback of a pre-recorded Assetto Corsa (Assetto Corso v.1.16, https://www.assettocorsa.it/en/, (accessed on 25 November 2022) KUNOS Simulazioni s.r.l., Via Degli Olmetti 39/B, 00060 Formello (RM), Italy) screen capture and were presented on the three lower monitors shown on the left side of Figure 4. During every session, the participant’s car was driving on a racetrack (Assetto Corso Tracks: https://www.assettocorsa.it/tracks/ (accessed on 25 November 2022)), with twelve other cars that followed the road. The selected racetracks contained curvy and straight street segments without two-way traffic to simulate one-way highways. For easier reference, we are using the notation: *k*-drive, where *k* indicates the number of actions the user needs to react to and set k∈{1,2,3} (Levels/Actions for reference: k=1: 
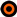

k=2: 
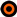
 / 
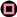

k=3: 
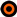
 / 
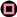
 / 
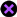
).

Similar to our dual-task *n*-back test (see Section 2.4), the subject is asked to solve a secondary task during level two and level three while observing the vehicle’s actions. For our secondary driving task, we ask the subject to search and add songs to a playlist using a mobile music application (Spotify App v8.6.12, 03/2021, Spotify AB, Regeringsgatan 19, SE-111 53 Stockholm, Sweden), that is presented on an iPad (iPad version 2021 Apple Inc., 1 Infinite Loop, Cupertino, CA 95014) next to the steering wheel. The songs that need to be added are shown on the top screen in our multi-monitor setup shown on the right side of Figure 4. To ensure a correct understanding we provide a detailed description of every task to the subject upfront and propose solution strategies, such as using the current race position, the presence of rearview mirrors or a tachometer. In addition to these instructions presented in advance, the subject will have at least one training session of 50 s before the test level starts, which can be repeated if that is requested by the subject. Similar to the *n*-back training sessions the positive and negative hits are reported after every training session. Before starting the driving experiments, the subject will run through three different baselines. During the first baseline, the subject will rest for five minutes and does not interact with the driving simulator, followed by a second baseline of two minutes, where the subject is observing the autonomously driving car in the test simulator without interaction or any imposed tasks. The third and last baseline phase (two minutes) is used to familiarize the subject with the music application, presented on the IPad. We run these baselines to ensure an ergonomic and familiar setting and record baselines for task-specific shifts, such as moving your eyes from the on-road screen to the iPad music application. We record positive (action was detected correctly) and negative hits (the wrong button was pressed, or no event occurred) as well as reaction time for our primary task. Additionally, we count the number of added songs that were added to the playlist for levels 2 and 3 for our dual-task experiment.

### 2.4. n-Back Test

In addition to our driving simulation, we conducted an *n*-back test [58] that is extensively used in the literature as a working memory paradigm. The experimentation scenario is inspired by [20]. In our single task schedule, we presented visuospatial stimuli as a blue square appearing at one of eight possible locations equally spaced and symmetrically around a white cross in the center on a black background. For dual-task tests, we additionally played recorded German consonants from the set {c, g, h, k, p, q, t, w} spoken by a male native speaker. Both versions are visualized in Figure 5. We repeated the test 6 times: 1-back to 3-back for visual-only stimuli and 1-back to 3-back for visual and audio-verbal stimuli, which were presented simultaneously for our dual-task workload paradigm. Prior to the complete *n*-back test run, the subject had different baseline phases. The schedule for the complete test run and the baseline phases is visualized in Figure 6. Each test was preceded by a short period of training that could be repeated on request by the subject, shown in Figure 7. Similar to Jaeggi et al. the stimuli were presented for 500 ms followed by an inter-stimulus-interval of 2500 ms [20]. The subjects were asked to react to repetitions using the buttons on the steering wheel. For a positive visual event, the subject was asked to press the 
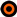
 and for an auditive event the 
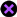
, respectively.

We recorded both, positive hits (when the reaction to a stimulus was correct) and negative hits (no stimulus was played, or the wrong button was pressed). For positive hits, we recorded the reaction time from the beginning of the stimuli until the button was pressed. The evaluation protocol for positive/negative hits and reaction time is described in Section 2.7. The subject received performance feedback (number of correct and incorrect hits) automatically after each training session, visualized on the monitor, to ensure that the task was understood correctly, before starting the actual test. The system feedback was formulated neutrally, while the supervisor did not give any feedback during the complete study, to reduce the effects of perceived social stress.

The appendix of this publication provides an overview of the experimentation sequence in Figure A2 of both conducted experiments.

### 2.5. Salivary Cortisol Responses

The measurement of glucocorticoids has proven to be a significant auxiliary value to detecting acute social stress responses using salivary samples [60,61]. Early studies identified an increase of cortisol, as a response to acute stress, transmitted by the HPA axis as a pathway to send signals to the organism. Cortisol is therefore a promising non-invasive biomarker for stress [62,63]. As our study does not impose any social evaluative thread but various levels of cognitive load, we are analyzing cortisol as a biomarker for our *k*-drive and *n*-back levels. Woody et al. have bisected the cognitive component and social evaluative component of stress and found that social evaluative thread in absence of cognitive load led to greater cortisol values while increased cognitive load had no main or additive effect [34] on cortisol. Contrary, Veltman and Gailard [64] have identified a high impact of different mental workload tasks and cortisol levels, while low task performance leads to higher cortisol levels. To capture the complete cortisol response [60,65], we measure four points in time for both tasks: Right before the test start (Pre-00), After finishing the test (Post-00), 10 min (Post-10) and 20 min (Post-20) after the test finishes.

### 2.6. Subjective Feedback

After every phase of the aforementioned *n*-back tests and drive scenarios, we ask the subject to self-evaluate and report the effort required to complete the given task. These subjective measures often vary. However, to a certain degree the self-reported ratings can reliably determine the CL put on a certain task [14,17,66]. In this work, we use the NASA-TLX, a well-established tool to analyze CL on a multidimensional, weighted scale [16]. We kept the dimensions as described by Hart [17]: mental, physical and temporal demand, performance, effort and frustration. The dimensions and associated questions were translated into German. Each dimension is also accompanied by a weighting factor through pairwise comparisons regarding their perceived importance. The final score is computed by a summation of the scores on each dimension times the corresponding weights (the number of times it was rated more important). The NASA-TLX score is then scaled to be between 0 and 100. In addition to the weighted version, we report the NASA-Raw Task Load Index (RTLX) without applying the weights. The overall score is computed using the average of all scores [15]. We collect three ratings for our single *n*-back test, three ratings for our dual workload *n*-back test and three ratings for our driving experiments. In addition, we evaluate perceived stress over the last month and therefore set a baseline for our evaluations. We utilized the Perceived Stress Scale (PSS) by Cohen et al. [67] to enable respondents to indicate how stressful their life is. Analog to the work by Cohen et al. [67] the 5-scale ratings (from “Never” to “Very Often”) are computed by summing scores of all items (including inversion for some questions). The respondents were able to express their positive and negative affect after the driving phase and the *n*-back test. Utilizing a 5-scale Positive and Negative Affect Schedule (PANAS) introduced in work by Watson et al. [68] with 10 positive feelings and 10 negative feelings, we report the affective state of the corresponding subject. The evaluation of affective states was conducted after both phases of this study. As PSS and PANAS questionnaires are used to measure a baseline stress level and affective states, but are not directly relate to specific phases of our experiment, we report the result in Appendix E.

### 2.7. Performance

Performance is commonly used to measure cognitive load as introduced in Section 1. Metrics to measure performance are highly task-dependent: We collect true positives (correct button pressed, TP), false positives (no event occurred, but the button is pressed, FP) and false negatives (the event occurred, but no input was recorded, FN) as well as reaction time for true positives. Both, for our *n*-back test, and our *k*-drive test, the maximal time between the beginning of an event and the time when the user could react to this event was set to 3 s. Similar to the nomenclature used in classification tasks we compute and report recall, precision and $F_{1}$-score. For this experiment we use:(1)$$\mathrm{precision}=\frac{TP}{TP+FP}$$
(2)$$\mathrm{recall}=\frac{TP}{TP+FN}$$
(3)$$F_{1}=2\cdot \frac{\mathrm{precision}\cdot \mathrm{recall}}{\mathrm{precision}+\mathrm{recall}}$$

The same metrics are also computed for the secondary auditive *n*-back task. For the secondary tasks during levels 2 and 3 of *k*-drive we only collect the correctly added songs to the playlist (false positives are not recorded) and report only our recall-motivated performance metric.

### 2.8. Physiological Features

For data acquisition, we used a Biopac MP160 system and specialized Biopac modules ECG100C, EGG100C, EMG100C, EDA100C, PPG100C, RSP100C and SKT100C for the corresponding modalities. We used a sampling frequency of 2000 Hz for all modalities, as storage was not an issue in this simulator setting and the analog synchronization across multiple devices, modalities and time stamps is ensured to be consistent. For ECG recordings, we use the Einthoven, Lead I and Lead II with a limp sensor placed below the shoulder and lower torso using *Ag-AgCl* electrodes (torso limp positions: right arm, left arm, left leg and right leg as neutral). The skin was prepared with an isopropyl solution. The surface EMG was recorded to capture the activation of the trapezius muscle using three electrodes. The reference electrode is placed on the seventh cervical vertebra (C7), and the other two electrodes for measuring the activation are placed at the right lateral trapezius muscle. The EDA electrodes were placed on the palmar side of the index and middle fingers (digitus manus 2 and 3) at the medial phalanges. To compensate for changes in conductance at the beginning of the recording, the placement of the electrodes was done at least 10 min before the recording started. For skin preparation, only warm water was used. The PPG signal was measured on the tip of the middle finger (digitus manus 3). Skin Temperature (SKT) was measured using a sensor placed on the tip of the index finger (digitus manus 2). All, EDA, PPG and SKT were measured on the non-dominant hand, which was placed on the leg and could rest without movement as no interaction was required. Respiration cycles were recorded using the BIOPAC MP3X respiratory effort transducer, measuring the change in thoracic circumference. During the early stages of our experimentation development, we recorded electroencephalographic recordings using a Mobita ConfiCab with 32 channels. We stopped the acquisition of these recordings during this experiment, as some subjects, reported headaches after the long duration of wearing the electrode cap.

We opted to extract features based on statistical signal properties and medical expert features for all biosignal modalities. Features extracted from electrocardiography recordings, reflecting the electrical activity of the human heart, are, except the heart rate itself, based on the concept of Heart Rate Variability (HRV). The HRV is impacted by modulations of biochemical processes, that correspond to the activation of the parasympathetic and sympathetic nervous systems. In addition to recordings of the electrical activity of the heart, we measure the mechanical activity using PPG sensors. Studies have extracted features based on the variability of the pulse waves, so call Pulse Rate Variability (PRV) features. In this publication, we are not interested in the estimation and computation of PRV features, because they have a strong correlation with HRV features, but are more sensitive to movement artifacts in our acquisition setup [69]. Recording PPG features as part of this database is still a valuable addition, as this modality enables the study of early fusion techniques reflecting measures like Pulse-Transit Time (PTT) [70] or detecting cognitive load using only wearable devices, sensing the activity of the heart using pulse waves in the extremities in the future. Based on Pham et al. [71], we solely use features reflecting the activity of the human heart on rhythmic scales and use the R-peaks for stable detection [72]. Our features are based on state-of-the-art techniques from the literature analyzing affective states, cognitive load, stress [33] and other biopsychosocial states. Our features are described in Table 1 and are among other techniques computed using time-based (RMSSD, SDNN, etc.) or frequency-based (HF, LF, etc.) methods. Another physiological response triggered by the Sympathetic Nervous System (SNS) is the activation of sweat glands. This electrodermal activity can be measured on the skin surface through conductance changes, that are caused by sweat and ionic permeability changes in and on the skin surface [73]. Current literature decomposes EDA signals into phasic and tonic components for the analysis of psycho-physiological responses [74]. We capture the tonic activation by computing the change in Skin Conductance Level (SCL) within a given interval and extract features of the faster-changing phasic component also interval-related by measuring Skin Conductance Response (SCR) and do not conduct any direct-stimuli related analysis of SCR events for our level-based experimentation setup. Both SCL and SCR based features are shown in Table 1. Past studies have also found a correlation between the activation of trapezius muscle activity using EMG recordings and cognitive load [75,76]. We have computed features based on the overall activity within a timeframe, e.g., by computing the root-mean-square of the preprocessed signal within a given interval, as well as features based on task-specific changes, such as the number of activations within that interval. To extract information from varying skin temperatures, that are associated with the cognitive load, we extract standard statistical features such as an increase or a decrease in temperature within a time window or mean and standard deviation and follow the methodology used in past studies, that found a correlation for both stress [33] and cognitive load [56]. We extract respiration features from our effort transducer values, by detecting lows (exhalation) and highs (inhalations) and use the amplitude difference between inhalation and exhalation, as well as the mean and standard deviation of respiration rates within a fixed window, as a measure for cognitive load [77]. All features used in this publication, are provided in Table 1. We compute all features with a rolling window with a predefined window size *W*.

### 2.9. Eye Tracking Features

Many studies have shown the relevance of eye-related characteristics for the prediction of a person’s cognitive state [85]. For eye tracking, we utilize the Tobii Pro Fusion (Tobii Pro Fusion I5S, Tobii AB, Karlsrovägen 2D, S-182 53 Danderyd, Sweden) eye tracker, a stereoscopic system with two eye-tracking cameras for the left and right eye, respectively, that records the subject’s eye movements and pupil size at a sampling rate of 250 Hz. We use the PyGaze library [86] to extract fixations (the activity of an eye looking at the same region for a certain duration), saccades (rapid eye movements between fixations), and blinks (defined by the absence of the signal for a certain time) from the acquired eye-tracking data. The fixation detection is based on a dispersion-duration method: if the dispersion of gaze data on the screen is below a certain threshold and has a duration of at least 100 ms, the data is considered a fixation. We select the dispersion threshold within an exploratory analysis as suggested by Salvucci et al. [87]. The duration threshold of 100 ms is a common choice [87,88,89]. Following Negi et al., we calculate fixations of three non-overlapping temporal ranges: 66–150 ms (short ambient), 300–500 ms (longer focal), and above 1000 ms (very long). This distinction allows us to gain additional insights into an individual’s conscious perception of objects [90]. Despite the strong dependence between pupil dilation and light reflexes, pupil size can provide information about the commitment of greater effort and the expectation of a difficult task [91,92]. High-frequency oscillations, also called hippus, can also indicate increased cognitive load [93].

In accordance with the literature, we extract multiple features based on fixations, saccades, blinks, and pupil dilation. An overview can be found in Table 2. For a window length *W* we extract the number and duration of fixations, saccades, and blinks as well as the mean saccade amplitude, thus the mean distance for a saccadic movement. As pupil-related features, we select the mean pupil size and two other features that take pupil change into account. To calculate the index of pupillary activity (IPA) we follow [93,94].

### 2.10. Videos

One of many indices in this study is the analysis of facial cues. These facial cues are sometimes closely related to messages, that express emotion, such as anger, fear, disgust, happiness, sadness or surprise. As we are highly interested in such behavioral expressions and their relationship to cognitive load, we record the subject’s face during the complete experiment. A BASLER acA1920 155 μm (Basler AG, An der Strusbek 60–62, 22926 Ahrensburg, Germany) camera was used to record a video stream with a resolution of 1920 × 1080 at 25 frames per second, which was triggered using the BIOPAC analog output, to ensure a fully synchronized recording. Depending on the position of the subject, we extracted 1024 × 1024 pixels as a region of interest and resampled it to a 512 × 512 frame. For action unit extraction we used the SVM model developed by Cheong et al. [95], using Deng et al. [96] as face detection model, Albiero et al. Img2Pose [97] as pose correction model and Chen et al. MobileFaceNets [98] to extracted landmarks. To mitigate effects, such as many-to-one correspondences of certain behaviors and emotions, or varying interpretations across subjects, we opted to use a sign-judgment-based action unit system instead. The Facial Action Coding System (FACS) system, introduced by Ekman and Friesen [99] and improved by Ekman and Rosenberg [100] specifies different action units. Martinez et al. present the current state of research and its application [101], while predicting cognitive load using visual facial cues has been studied in Li and Busso [102], Yuce et al. [29] and Viegas et al. [28], on which we base our decision to extract the action units presented in Table 3. For our statistical analysis and machine learning experiments, we count the frames with active actions unit within the given window and denote the number of action units similar to our physiological features with # as count.

### 2.11. Preprocessing, Artifacts and Data Collection

The recording of biosignals is prone to various artifacts, while we designed our recording environment with great care, in some situations it is not evitable that artifacts degrade signal quality. Due to the complex nature of our environment, we were not able to prevent 50 Hz power line noise and removed this frequency component, if applicable, and all higher harmonics from the recorded signals using a notch filter. For our ECG feature extraction pipeline it is crucial to detect the R-peaks with high sensitivity and specificity. To ensure only sequences with sufficient quality are used during evaluation, we have computed Signal Quality Index (SQI)’s based on work by Zaho and Zhang of both leads and selected the lead with better SQI’s or if both leads are corrupted, we exclude those intervals from the evaluation [104]. For video, we removed frames from evaluation if the subject’s head moved outside the recording region. For eye tracker measurements we removed samples with a Tobii Fusion Pro system code indicating the detection of both eyes. For all other modalities, our artifact rejection was integrated by information from the examiner that noted events to the protocol during acquisition. These time intervals were manually removed. Otherwise, we computed features over a two-minute rolling window with a step size of five seconds of our continuous signal. We employed subject-wise z-score normalization (subtracted mean and scaled to standard deviation) using every feature computed within the complete phase of our *k*-drive and *n*-back test, including baselines, training and testing phases. For statistical evaluation and machine learning, we used the window that was in the center of the respective phase described in Section 2.3 and Section 2.4 and removed outliers using the Median Absolute Deviation (MAD)-rule [105].

To summarize our feature-extraction and preprocessing protocol, for all records used in our statistical evaluation and machine learning pipeline, we start by rejecting artifacts as described above in this section and exclude corrupted parts of the signal. If the examiner reported recording issues during acquisition (e.g., ECG lead fall-off), we removed the corrupted parts from the sequence. Given the recorded, fully synchronized trigger signals, we define the windows of the baselines and levels. Following that, we computed all features for each modality using a rolling window over the complete experiments of *n*-back and *k*-drive and normalize the extracted trials using subject-wise z-score normalization. We used the MAD-rule to remove outliers of our features. When working with combinations of multiple features, we drop the complete instance (feature vector) that contains a missing value from our dataset and therefore out machine learning experiments. We do not employ any imputation technique, for missing data, but remove the entire sample that contains a missing feature value. We report the percentages of available records after these steps in Appendix H for reference.

### 2.12. Statistical Evaluation

To analyze the effect of different experimentation levels (factors), on our extracted expert features (values) we conduct a one-way Analysis of Variance (ANOVA) for repeated measurements, observing if there are significant changes. We determine if the data is distributed normally, using a Shapiro–Wilk test. If the criterion of normality for ANOVA is not met, we conduct a non-parametric Friedman test as omnibus test instead. The assumption of sphericity was controlled by a Mauchly’s test. Whenever necessary, we employ a Greenhouse-Geisser correction. As we conduct multiple comparisons we adjust our *p*-values using Holm–Bonferroni correction for our experiments (*k*-drive and *n*-back) separately. For features that changed significantly according to our repeated measures ANOVA, we conducted post hoc-tests. If the post hoc-test results show a *p*-value below our level of significance α=0.05 we report if the mean value is increasing or decreasing for that specific feature. If the conditions for normality are met, we use a two-sided paired t-test, otherwise, we use the non-parametric Wilcoxon test as a post hoc-test. These tests are executed for our features described in Section 2.8, Section 2.9 and Section 2.10.

### 2.13. Classification Tasks and Evaluation Protocol

**A**utonomous **D**riving Cognitive Load **A**ssessment Data**base** (*ADABase*) was recorded to study subjects under different levels of cognitive load while observing an autonomously driving vehicle, using multi-modal representations of physiological-, behavioral-, subjective- and performance-based measurements. We create various downstream tasks for classification and regression. As a first task, we separate our recording into two levels of cognitive load:A *Low Load Class*, without any task or undemanding tasks imposed on the subject.A *High Load Class* with tasks of high demand introducing high cognitive load.
Using this setup we evaluate three different experiments: For the visual-only *n*-back task we select the observation-only-baseline and the 1-back level as low load class and the 2-back and 3-back levels as high load class. The same phases are used for audiovisual classification. For complete *n*-back classification we used both baselines (resting and no-task observation), 1-back visual-only, 1-back audiovisual testing phase as low load class and 2-back and 3-back visual-only and audiovisual for high load. Using this classification scheme has several advantages of having a balanced class distribution, being able to evaluate single-only and dual-only vs. single-and-dual task loads, and capturing different levels of the load imposed by the given tasks. We propose to use this labeling scheme for *k*-drive, too. For our low load class, we use the baseline with an observation-only driving example as low load class and 1-drive as high load class. For our dual-task-load classification of *k*-drive, we use the baseline with observation-only driving and the baseline with secondary-task observation only as low load and 2 and 3-drive dual task as high load. We also train a classifier on the combination of *n*-back and *k*-drive schemes, leading to a single-task, dual-task and a combination of both binary classifications. It is important to note that these classification groups have been carefully selected to include similar tasks (e.g., only observation of *n*-back tests in baseline instead of simple rest periods) to mitigate the effects of distribution shifts, like not looking to the monitor or infotainment system and light patterns that influence the pupil dilation.

In addition to this very coarse-grained experiment of low vs. high load [48], we create an experiment with three classification levels, low-load, medium-load and high-load. We do not differentiate between a single and a dual classification task but study the combination of both tasks for *n*-back and *k*-drive. For *k*-drive, we use the observing-only-baseline and 1-drive for low load, 2-drive for medium load and 3-drive for high load. For *n*-back we use the observation-only baseline and visual-only 1-back as low load, 2-back visual only and 1-back dual task for medium workload and 3-back visual-only and 2-back and 3-back audiovisual for high load. It is noteworthy that the records collected during the specific phases, using these splits are not equally distributed.

We propose several targets for our regression tasks. The first task is to use performance metrics only, averaging a score based on the recall of the primary task (scaled between 0 and 1, where 0 is the best score (highest recall) and 1 is the worst of the complete study population) and the reaction time of the primary task (scaled to be between 0 and 1, where 0 is an instant reaction and 1 is a reaction within 3 s as the maximal possible reaction time). As a second target, we use the NASA-RTLX score, scaled between 0 and 1, where 1 is the highest task load reported by the specific subject and 0 is the lowest reported task load of that subject. We drop all baselines from this dataset. For our third task, we compute the average of both targets and therefore use performance and subjective rating as a continuous target between 0 and 1 for our regression algorithms.

For evaluation, we conduct several experiments. We group our features modality-wise and train a classifier using the binary-task-classification to train and evaluate classifiers for all sub-tasks: single-*n*-back, dual-*n*-back, single-*k*-drive, dual-*k*-drive the combinations of both, single and dual-task load, and both experiments, *n*-back and *k*-drive. Using this experimentation schedule we can as an extension to our statistical evaluation in Section 2.12 determine the predictive power of unimodal and multimodal models.

### 2.14. Machine Learning Algorithms and Training

Past studies in the affective sensing multimodal human sensing community have shown promising results using eXtreme Gradient Boosting (XGBoost) as a machine learning technique for classification [106]. We follow their finding of using XGBoost and verify this approach, by training and evaluating a Support Vector Machine (SVM) with a linear and a radial kernel, as well as a k-Nearest-Neighbors (kNN) classifier using data from our experiment. To prevent data leakage in our machine learning pipeline, we use nested k-fold cross-validation with 10 inner folds as the validation set and 10 outer folds as the testing set [107], taking bias and variance of our models into account [108] and therefore provide reliable results. The data is split subject-wise. For our binary-classification task, we report Area Under the Receiver Operating Characteristic Curve (AUC) and F1-score. For our three-level classification task, we provide a confusion matrix. The predictive power of our regression task is expressed as $R^{2}$ and Mean Squared Error (MSE). For hyperparameter optimization, we run a tree-structured parzen estimator on the validation of every inner fold [109] and optimize for maximal AUC for our classification tasks and maximize $R^{2}$ for our regression task. We optimize for the number of estimators, depth, lambda and learning rate for XGBoost, gamma and C for radial basis SVM, C for linear SVM and neighbors and weight function for kNN. As input for our classification, we use the same windows used for our statistical tests, described in Section 2.12 and Section 2.11. For training, validation and testing we use feature windows only recorded within the phases described in Section 2.13. In addition to our experiment, using different groups of unimodal and multimodal inputs, we report the feature importance of our XGBoost classifier to explain our model behavior. We use gain, weight and coverage for every feature and report the importance of the top-5 features for every modality.

## 3. Results

We structure our results similar to the sections in Section 2: After giving an overview of the recorded data in Section 3.1, we present the results of the statistical evaluation for subjective measurements in Section 3.2, for performance measurements in Section 3.3, for all physiological signals for *n*-back and *k*-drive in Section 3.4, Section 3.5 and Section 3.6. Behavioral statistical results are presented in Section 3.7. The personality traits of our cohort are presented in Section 3.8. We visualize our features representations using t-SNE in Section 3.9. The results of our machine learning experiments are presented in Section 3.10.

### 3.1. ADABase and Timescales

Each subject participated in both experiments described in Section 2.3 and Section 2.4 in random order, resulting in 25 subjects with *n*-back as the first experiment and 26 subjects with *k*-drive as the first experiment. Due to the malfunction of the eye-tracking hardware, for six subjects the eye-tracker data is missing. A defect in the camera trigger line led to synchronization issues of video data with the rest of the test system and some subjects refused the recording of video signals with privacy concerns. We excluded the video data for 17 subjects, leading to missing action units for these subjects. For two subjects the skin temperature data saturated at maximum, caused by defective sensors connectors. For some subjects Eindhoven lead II of the ECG was noisy or the electrode fell off. As an alternative, we used the other lead for R-peak detection and feature computation. If temporary errors occurred during the recording (e.g., the subjects moved their face during the questionnaire phase outside the recording area), we drop only these specific intervals using our artifact correction mechanisms described in Section 2.11.

Figure 8 visualizes features computed for one session of a sample subject during both experiments. The color-coded background data highlights the phases on different timescales. Every test phase is proceeded by at least one training phase, for all tests of n∈{1,2,3}-back for visual-only and audiovisual task loads, 1-drive for single task load settings and k∈{2,3}-drive dual task loads. The *n*-back baseline 1, and *k*-drive baseline 1 were resting phases of at least 5 min, where the subject was instructed to rest without any interference or additional tasks. During the second *n*-back baseline phase the subject observed an *n*-back sequence without any task imposed on the subject. In the second *k*-drive baseline an autonomous driving simulation was presented without any task imposed on the subject. The third *k*-drive baseline was introduced to the iPad infotainment application, used for dual-task load scenarios of *k*-drive. As introduced in Section 2.3 and Section 2.4 these baselines enable the study of distribution shifts in future work. Salivary Cortisol, NASA-TLX and recall for the primary and secondary tasks are recorded after test phases or in the case of salivary biomarkers at time points when the test schedule required it. In addition to the fine-grained phases used in this publication, we would like to highlight the ability of our published *ADABase* to learn representations of cognitive load on different timescales. Starting with a continuous prediction, only limited by physiological processes in the test subject, followed by more medium-grained task-specific phases, that combine training and testing, to coarse-grained phases that differentiate between single and dual task loads tasks.

### 3.2. Subjective Evaluation

After every testing phase, we asked the subject to answer a NASA-TLX questionnaire. We conducted t-tests of raw NASA-TLX values between 1-back and 2-back, 2-back and 3-back and 1-back and 3-back and the same for all levels of *k*-drive. All Holm-adjusted *p*-values were with p<0.001 below the level of significance α=0.05 to reject the null hypothesis (two-sided mean alternative) and show a significant change across the levels conducted in this study. Figure 9 shows the means of all six TLX dimensions during different phases of the experiments and Figure 10 shows boxplots of weighted and unweighted (raw) TLX values and the raw ratings for every dimension and level.

### 3.3. Performance

For every level in our single and dual task *n*-back tests and all our *k*-drive tests, we conducted t-tests for both recall and precision of the visual performance between the first level and the second level, the second level and the third level, as well the first and the third level. All Holm-adjusted *p*-values are smaller than 0.001. We report primary and secondary task performance and reaction time for all tests in Figure 11.

### 3.4. Biomarkers

We report the relative median values and quantiles of the raw non-normalized cortisol measurements in Figure 12. The null hypothesis of equal means before starting the experiment (Pre-00) and right after each levels of *n*-back (single and dual) and *k*-drive (Post-00) could not be rejected with a sufficient level of significance. This result is in line with other studies analyzing cortisol after phases of high cognitive load [34]. In addition to the reported relative values of *n*-back and *k*-drive tests, we report the cortisol values over the absolute daytime, showing the circadian variation of cortisol of the sample population.

### 3.5. Physiological Features—*n*-Back

For statistical evaluation of features measured during *n*-back (*n*-back: Section 2.4, features: Section 2.8) levels, we are using the evaluation protocol described in Section 2.12. We analyze our features extracted from the biosignal modalities and the eye tracker data in Table 4. The results of the omnibus test showed a significant change for features extracted from ECG, EMG, PPG, skin temperature, respiration and eye tracking data during resting baseline, observation only baseline and all *n*-back tests with single- and dual-task load. Comparing the features during the observation only baseline and the medium level 2-back and high level 3-back cognitive load task as post hoc experiment and evaluation of an increasing or decreasing mean for significant changes shows an increased heart rate, a decreased heart rate variability, a decreasing skin temperature, decreasing mean respiration rate and an increasing IPA. For the EDA activity we could not reject the null hypothesis for all repeated measures. We selected the baseline tutorial phase for our post hoc test because no cognitive load was introduced during this phase, while the subject was still observing an *n*-back sequence with similar lighting and pose conditions. All results are reported in Table 4.

### 3.6. Physiological Features—*k*-Drive

We evaluate our measurements of features during all levels of *k*-drive following our statistical evaluation protocol desribed in Section 2.12. Similar to our statistical evaluation in Section 3.5 we see a change for features extracted from ECG, EMG, skin temperature, respiration and eye tracker data using our ANOVA for repeated measure evaluation protocol described in Section 2.12. In contrast, the PPG showed no significance, while the EDA showed a significant change. For our omnibus tests, we used all three baselines described in Section 2.3 and all levels k∈{1,2,3}-drive. The post hoc test using the second baseline with driving simulation observations only and all three levels shows similar to the statistical evaluation of the *n*-back experiment an increasing heart rate, a decreasing heart rate variability, a decreasing skin temperature and an increasing IPA. All results are reported in Table 5.

### 3.7. Behavioral Features—*n*-Back and *k*-Drive

For both experiments *n*-back and *k*-drive, we extract action units from facial videos as described in Section 2.10. We evaluated the computed features using the same statistical setup as described in Section 3.5 and Section 3.7 and report the results for our *n*-back test in Table 6 and Table 7. The statistical evaluation of our behavioral features showed a significant change in the number of activations of outer brow raiser (AU2), chin raiser (AU17), lip tightener and pressor (AU23 and AU24) during *n*-back and *k*-drive for our omnibus test. Our *n*-back tests change the number of activations of brow lowerer (AU4), cheek raiser (AU6), lip corner depressor (AU15) and closed eyes (AU43). The *k*-drive tests showed a significant change in inner brow raiser (AU1), upper lid raiser (AU5), lip corner puller (AU12), dimpler (AU14), lip strecher (AU20) and jaw drops (AU26).

### 3.8. Personality Traits

When characterizing stress responses using multimodal physiological data, personality traits play an important role [61]. Figure 13 reports openness, conscientiousness, extraversion, agreeableness and neuroticism of our study population. As an additional test, we computed the spearman correlation between the mean overall NASA-TLX score of all levels for one subject and the reported personality and found r=−0.401, CI=[−0.61,−0.14] for openness, r=0.053,CI=[−0.23,0.32] for conscientiousness, r=−0.082, CI=[−0.35,0.2] for extraversion, r=0.164,CI=[−0.12,0.42] for agreeableness and r=−0.019, CI=[−0.29,0.26] for neuroticism, with *r* as correlation coefficient and CI as 95% confidence intervals around *r*.

### 3.9. Representations

Using our extracted biosignal features, we have computed t-SNE representations with two components of different feature subsets. Representations are computed for all participating subjects. The colors indicate the coarse-grained binary classes, where blue contains the ground-truth labels *n*-back baselines, *k*-drive baselines, 1-back single task and level 1 drive task and red indicate phases with high cognitive load: 2/3-back single task, n∈{1,2,3}-back dual-task load and 2/3-drive tests. In other words: red indicates a high cognitive load, while blue indicates a low or no cognitive load. The last plot shows as a color map the regression target described in Section 2.13, with a darker blue for higher scores and a lighter green for lower scores. The separation of clusters visualized in Figure 14 as 2d t-SNE components of our high dimensional feature vector, described in Section 2.8 are a promising indication of useful representations for classification tasks in Section 3.10.

### 3.10. Machine Learning

The automatic detection of cognitive load using computerized programs is a key ingredient for deployment in semi-autonomously driving cars. We conduct several experiments described in Section 2.13 using the machine learning pipeline described in Section 2.14 and *XGBoost* as model. Our first experiment determines the effect of using features from different and multiple modalities as input for binary task classification. The results are reported in tabular form in Table 8. In addition, we train the models on various subsets of our collected data, containing single- and/or dual-task load *n*-back tests, single- and/or dual-task load *k*-drive tests and combinations of both. The AUC of models trained on the combination of all biosignal features is 0.81 ± 0.05, for the combinations of all *n*-back and *k*-drive tests. Models using only PPG as input achieve a AUC of 0.55 ± 0.06, using only respiration 0.65 ± 0.06, using only the trapezius activity 0.67 ± 0.06, using the electrical activity of the heart 0.75 ± 0.06 and models using the electrodermal activity achieve a AUC 0.68 ± 0.04. Using exclusively features based on action units data lead to a AUC of 0.69 ± 0.07. Models using solely features based on eye tracker data achieved a AUC of 0.84 ± 0.05. Models trained and evaluated on the combination of features from biosignals, action units and eye tracker data outperform unimodal features or other combinations with a AUC of 0.90 ± 0.04. This finding holds for single-and-dual-task *n*-back and *k*-drive tests. Another noteworthy finding is that eye tracker features perform equally well for all conducted tasks with a AUC of 0.86 ± 0.04 for single-and-dual-task *n*-back test and a AUC of 0.89 ± 0.06 for single-and-dual-task *k*-drive test, while other modalities such as EDA perform well for *k*-drive with 0.88 ± 0.03 but have a dropping performance for models trained and evaluated on single-and-dual task *n*-back tests with a AUC of 0.71 ± 0.08.

To further study the impact of all acquired features on classification performance for our binary classification task, we analyze the models that used all tasks, levels and features from all modalities for training. As described in Section 2.14 we use *XGBoost* for classification and can therefore express the feature contribution to the final classification as *gain* metric, that implies the relative contribution of the corresponding features to the final prediction of the model. Figure 15 shows the gain importance of the top features for each modality used in our classification task, showing contributions form IPA and $\mu_{PS}$, as high contributing features from the eye tracker modality, the change of skin conductance within a time interval as a feature with high gain in our *XGBoost* model and both heart rate and heart rate variability based features extracted from the electrical activity of the heart. In Figure A4, we present a supplementary analysis of the feature importance of our trained models.

In addition to our low vs. high cognitive load task, we introduced a classification task for multiple intensities of cognitive load in Section 2.13. Table 9 visualizes confusion matrices across three different sets of features, using only biosignals, exclusively utilizing eye tracker-based features and a combination of eye tracker-based features, biosignals and action units. It becomes apparent that the detection performance drops significantly compared to simple binary classification only differentiating between low and high cognitive load, while low-load is still detected in 92 percent correctly for *k*-drive using all features, high-load is often confused with medium-load and only detects 69 percent correctly. This holds for different subsets of features and tasks. We can observe that the identification of multiple levels, degrades performance, leading to an F1-score of 0.72 ± 0.09 for using all features for the *k*-drive test.

Previous literature found that *XGBoost* models performed best for the detection of drivers’ distraction from physiological and visual signals [106]. We confirm these results by training a SVM with linear and radial basis functions and a kNN classifier. All results are reported in Table 10.

In addition to our classification tasks, we train a regression model using a *XGBoost* regressor. The regression targets are described in Section 2.13. Using only performance-based targets, we were able to train a model with an $R^{2}$ score of 0.51 ± 0.07, using subjective ratings acquired using NASA-TLX scores, we get 0.48 ± 0.08, and for a linear combination (average of both), we achieve a $R^{2}$ score of 0.54 ± 0.14. All results are presented in Table 11.

## 4. Discussion

The presented results compiled in the previous chapter of this study provide a valuable addition to the research community assessing the effect of cognitive load using multimodal measurements.

With a carefully designed study population and a detailed cohort description, we conducted two experiments with various levels of cognitive load. As a reference experiment, that is comparable to experiments on cognitive load in related work, we asked the subjects to participate in a single- and dual-task load *n*-back test with visual-only stimuli and audiovisual stimuli inducing cognitive load. As an application-motivated test, we developed the *k*-drive test, which is inspired by the observations of semi-autonomously driving cars with only a few driver-car interactions required. This test imposes cognitive load by an increasing number of events the subject needs to respond to. In addition, a secondary task of interacting with a car infotainment system needs to be executed for dual-task load tests.

Our study compiled a recording setup for multiple modalities. Our *physiological* measurements include biosignals such as ECG, PPG, EDA, EMG, skin temperature and respiration recordings as well as eye tracker data. We found significant changes in established expert and statistical features during both, *n*-back and *k*-drive, which revealed an increasing heart rate, decreasing heart rate variability, an increasing number of peaks in the electrodermal activity, a decreasing mean skin temperature, a decreasing respiration rate and an increasing eye tracker IPA during cognitive load compared to non-load baselines, to name only a few. Our findings of a positive correlation between IPA and cognitive load align with previous research that has reported promising results using IPA in cognitive load measurement [93,110,111]. Additionally, our omnibus test has shown the statistical significance of fixation features which also prove to be relevant in cognitive load prediction. This finding complies with literature [30,85,112,113,114] In addition, we recorded *behavioral* measurements based on facial videos. Using simple features based on the number of activated action units within a certain time frame, we found several action units that changed significantly for increasing levels of task demand of *n*-back and *k*-drive levels. These observations are supported by the findings of Yuce et al. who analyzed the link between action units and cognitive load while driving [29]. We measured several *performance* metrics for all conducted *n*-back and *k*-drive tests and levels for both secondary and primary task loads, including precision and recall. In combination with the *subjective* feedback, acquired through NASA-TLX questionnaires following every test, we found properties of human responses, that are statistically significant across all analyzed dimensions: *physiological*, *behavioral*, *performance* and *subjective* measurements. Comparing our collected dataset with work analyzed by a recent review of related data collections by Seitz et al., confirms the validity of our multimodal approach for the computerized detection of cognitive load, differentiating our setup with a combination of multiple modalities, that all have been used in subsets, but not as a combination during the same protocol [115]. The review of Seitz et al. further helps to highlight our contributions, as we provide multiple potential targets for model training and evaluation: subjective ratings, performance-based measurements and task-specific level information, which all have been studied independently in related work and are presented as alternatives in this work [115].

Starting with a unimodal input we train several machine learning models and present the predictive power of each biosignal modality, eye tracker data and facial action units separating low and high levels of cognitive load. By combining features of various modalities, we present the predictive power of all biosignals, biosignals and eye tracker data and the combination of eye tracker data, biosignal and action units, leading to an overall AUC, using the complete dataset consisting of collected single- and dual-task load n∈{1,2,3}-back and k∈{1,2,3}-drive test of 0.91 ± 0.02. In general, we can observe that combining modalities leads to an improvement in classification accuracy. This is especially evident in the case of using data from both tasks and emphasizes the importance of a multimodal approach, given that other publications report similar results [116,117,118]. In addition, we report the gain feature importance values of our *XGBoost* classifier, showing important contributions of features like heart rate, heart rate variability, change of skin conductance within a given time interval, IPA, fixations and saccades of our eye tracker data as features with strong predictive power in our binary low-vs-high cognitive load classification task. During *n*-back we were able to control the lighting conditions to be more stable, compared to the application-motivated *k*-drive test. This could be a reason, why IPA is the most important feature for the detection of cognitive load for one task while losing importance for the other task. Nonetheless, we found eye-tracking-based features to have a high contribution (see Appendix G) for the detection of cognitive load in this publication. This finding is in line with work by Ahmed et al. [119], who observed, that pupil-related features have the highest feature importance in comparison to ECG and respiration for multimodal classification. We evaluate several machine learning techniques using kNN, SVM with linear and radial kernel and find that XGBoost performs best using our complete set of features. We introduced a new task separating three levels of cognitive load: low, medium and high and found a high true positive rate for the detection of tasks that imposed only low loads onto the subject. The separation of levels with medium and high cognitive load leaves room for improvement in future work. It has also been observed in other publications that separation with more than 2 classes is challenging [56,117]. One possible reason for this could be a strong inter-subject variability, making fine-grained classification difficult.

In addition to our classification tasks, that separate the different levels of CL imposed onto the subject based on the task difficulty of each level, we introduced a regression task, that used *performance* metrics, *subjective* ratings and the linear combination of both measures as targets, resulting in a $R^{2}$ score of 0.54 ± 0.14 for all levels and tasks. To summarize our work, this work contributes:The introduction of *k*-drive, a novel and close-to-real-world autonomous driving task to study cognitive load.The collection of *physiological*, *behavioral*, *subjective* and *performance* measurement from 51 subjects while participating in levels of increasing task difficulty with single and dual workload scenarios.The extraction and evaluation of features from ECG, PPG, EMG, EDA, respiration belt, skin temperature, eye tracker data, facial video recordings using a detailed statistical evaluation protocol.The validation of the collected data using statistical methods before training and testing of machine learning models.The training and evaluation of machine learning models using unimodal and multimodal inputs for cognitive load estimation.The analysis of feature importance for models trained with multiple inputs.The introduction of novel machine learning tasks and training of baseline models as reference for future contributions by the affective sensing and machine learning community.
Accompanying this publication we release a subset of 30 subjects, containing the recorded and unprocessed sensor data, questionnaires, performance metrics and NASA-TLX scores, enabling the research community to develop and evaluate novel algorithms for cognitive load estimation. We encourage the research community to build novel algorithms and machine-learning methods using the released data.

Our future work will include the analysis of distribution shifts of machine learning models that are trained on one task and evaluated on another task. An important next step before deploying these algorithms in the wild is the analysis of the robustness and stability of fusion techniques for multimodal machine learning when one or more modalities are corrupted or perturbed. In direct line with this work will be the study of models trained in an end-to-end manner such as deep neural networks using the raw input signals for CL prediction. Another important factor is the inclusion of the reported personality traits as input to machine learning models to improve model performances using personalized information. In this publication, we have selected a constant window length of two minutes for all features and all inputs, based on results from literature introduced in Section 2.8. However, physiological responses manifest themselves on different timescales and the optimization of both latency and window length of all extracted features during or after phases with high CL may further improve the detection rate using computerized machine learning models. Given the precise measurements of time with a resolution of only 0.5 ms for events and phases of different levels of CL and the results of this study this is a promising direction that will be explored and described in future work.

## Figures and Tables

**Figure 1 sensors-23-00340-f001:**
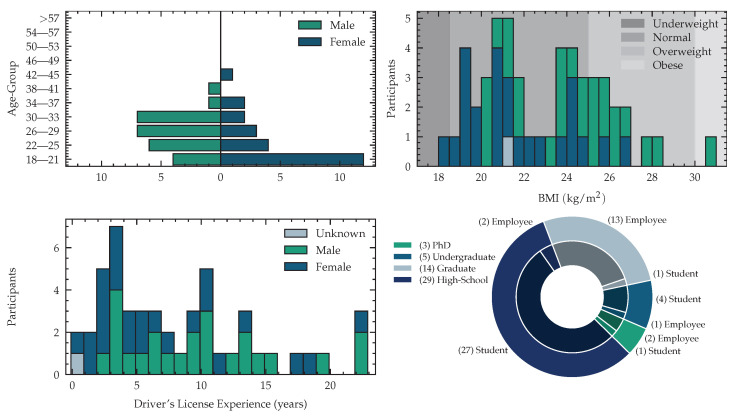
Demographic data was answered by subjects before participation. (**Upper-Left**): Pyramid with age groups over several subjects within that group (One subject did not answer the age and sex questionnaire). (**Upper-Right**): Histogram of BMI with highlighted classes according to WHO healthy lifestyle classification guidelines [55]. (**Lower-Left**): Histogram of driver experience in years. Subjects with no driver’s license have been set to zero years. (**Lower-Right**): Pie-Chart with current occupation (student, employee) within different formal education (high-school, undergraduate, graduate, Ph.D.). Inner-Circle: Occupation, Outer-Circle: Highest Obtained Degree.

**Figure 2 sensors-23-00340-f002:**
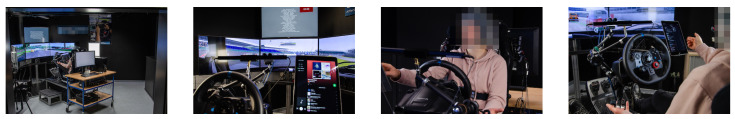
(**Left**)-to-(**Right**): Overview of the test environment with a depicted multi-monitor setup, subject in driving seat and acquisition setup; View from the subject’s right shoulder perspective; View over the shoulder, with the steering wheel, camera, eye tracker and tablet cockpit; Frontal view of the subject fully connected to all bio-signals.

**Figure 3 sensors-23-00340-f003:**
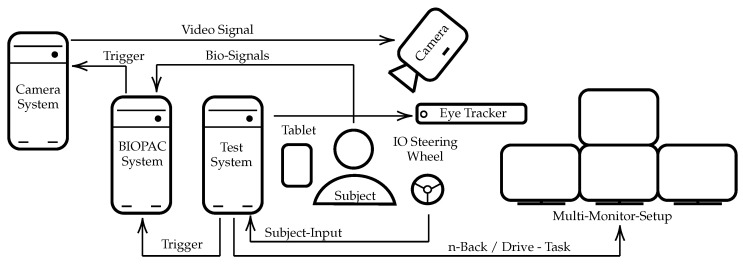
System components used during both experiments. Synchronization over multiple components mitigates effects such as clock drifts between different computer systems. The main test computer runs the experimentation software written in PsychoPy. The subject uses the buttons on a steering wheel as an input device during various experiments. The eye-tracker is synchronized with the main test computer. The camera system is triggered (on rising edge) with a constant low/high analog trigger signal having a frequency of 25 frames sent by the Biopac system. The camera is set to idle and automatically starts when the Biopac system is started at the beginning of the experiment and automatically stops as soon as the Biopac software stops. The Biopac system and the main test system are connected through a TTL line, sending the current state of the experiment at times of experiment phase transition as 255 bit-encoded signal, all interactions with buttons, as well as a continuous signal for synchronization.

**Figure 4 sensors-23-00340-f004:**
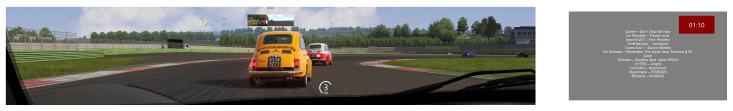
Screenshot of experimentation setup during driving. The left image shows the split-screen visualized on the three monitors with driving instruments, such as position, back mirror, current gear and velocity. The right image shows the remaining time during the experiment, as well as a list of songs the subject is instructed to add to the playlist on the virtual tablet cockpit.

**Figure 5 sensors-23-00340-f005:**
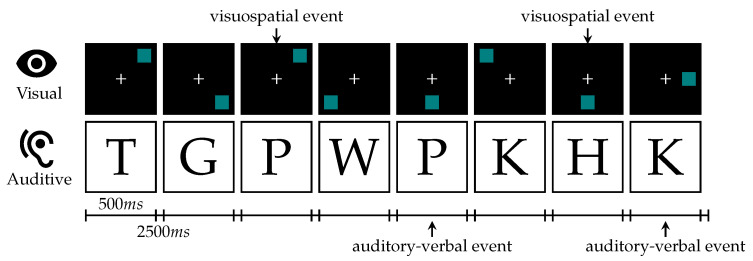
Example of visuospatial and auditory-verbal stimuli, presented during dual *n*-back test with level 2 (dual-task 2-back test) [20]. Stimuli are presented for 500 ms followed by an inter-stimulus-interval of 2500 ms. Example with two visuospatial and two auditory-verbal repetitions. The expected reactions are presented by arrows below and above the figure.

**Figure 6 sensors-23-00340-f006:**
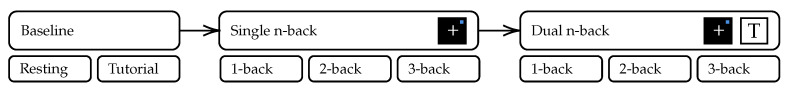
Visualization of the *n*-back test protocol. Baseline measurement consists of a resting phase of 5 min, followed by a 2 min tutorial, that shows stimuli and explains input controls to the subject. Two distinct phases of single and dual task *n*-back tests for n∈{1,2,3} are presented to the subject.

**Figure 7 sensors-23-00340-f007:**
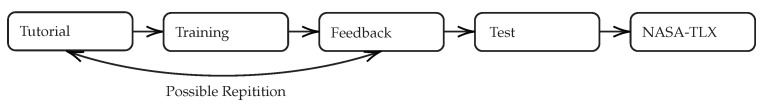
Protocol of a single *n*-back test sequence. The subject is able to repeat the tutorial and training phases. After training completion, the achieved score is reported to the participant (number of hits and false alarms are shown). Each test run is completed with a NASA-TLX assessment. The test phase takes 2 min to complete.

**Figure 8 sensors-23-00340-f008:**
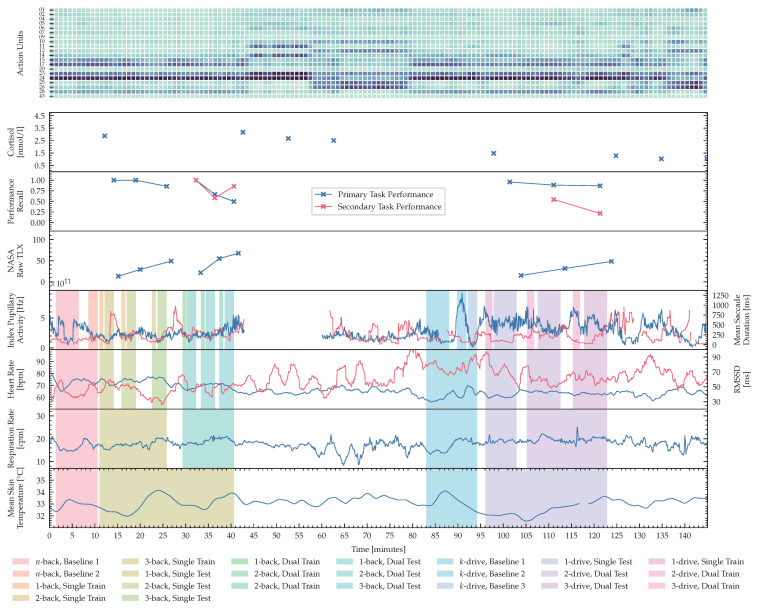
Visualization of the complete recording session for a single subject. Starting at the top-row: Continuously computed action units sampled with 25 fps, that are summed over 1-minute intervals and visualized as colors by occurrence (darker colors show higher activation). Followed by cortisol measurements, performance ratings and subjective feedback as NASA-RTLX scores. The last four rows visualize computed features with a rolling window of 2 min and step-size of 5 s, from eye-tracker, ECG (blue: HR, red: RMSSD), respiration and skin temperature measurements. Aligned with the overall time axis the different phases are presented as background colors. As described in Section 2.2, Section 2.3 and Section 2.4 the experiments (*n*-back and *k*-drive) are in randomized order. In this example, the *n*-back test was conducted first, followed by the questionnaire phase, and the driving phase. The *n*-back protocol was conducted as described in Figure 6 and Figure 7: Single task *n*-back first in three levels, followed by dual-task *n*-back in three levels with training and testing phase. This subject did not repeat any training. If obvious measurement artifacts occurred during acquisition, the corresponding features were removed for downstream evaluations. This example shows corruption in the features of the eye tracker, as the subject’s head moved outside the recording region of our eye tracker during the phase between *n*-back and *k*-drive.

**Figure 9 sensors-23-00340-f009:**
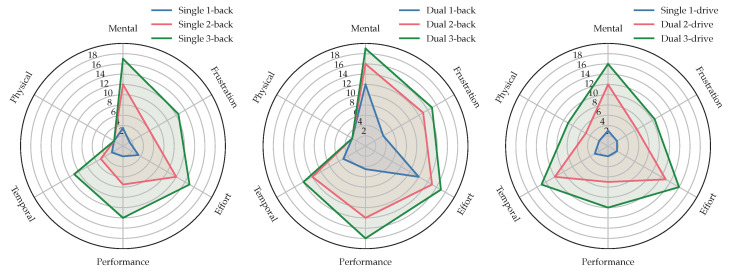
The three plots summarize the NASA-TLX answers of all six dimensions as the mean over all subjects. From left to right: Mean answers during single *n*-back, dual *n*-back and *k*-drive tasks. Reported numbers are not weighted and reflect the raw task load index described in Section 2.6. This radar visualization shows that mental, frustration, effort, performance and temporal dimensions increase with cognitive load tasks of higher intensities (n∈{1,2,3}-back) and dual vs. single, while the change in the subjectively perceived physical load is, compared to these dimensions, smaller. For *k*-drive of dual and single tasks, every NASA-RTLX dimension increases with a higher CL.

**Figure 10 sensors-23-00340-f010:**
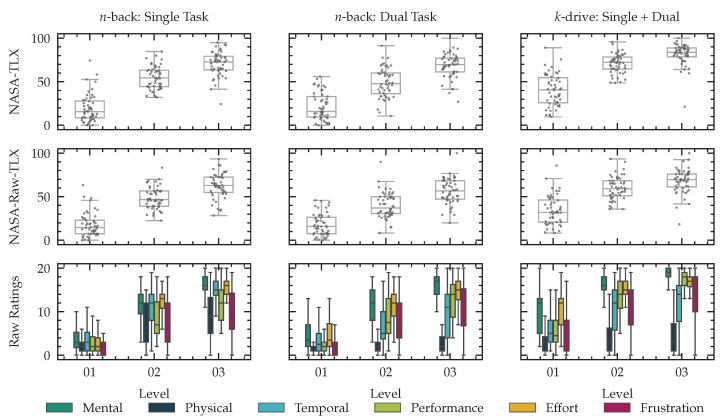
Results of subjective feedback for *n*-back and *k*-drive tests of increasing intensities of cognitive load. The first row shows the weighted NASA-TLX ratings during visual-only stimuli *n*-back, audiovisual *n*-back stimuli and single-task-only 1-drive and dual-task 2-drive and 3-drive tests. The second row shows the unweighted raw-TLX scores. The last row visualized the raw rating for every of the six NASA-TLX dimensions.

**Figure 11 sensors-23-00340-f011:**
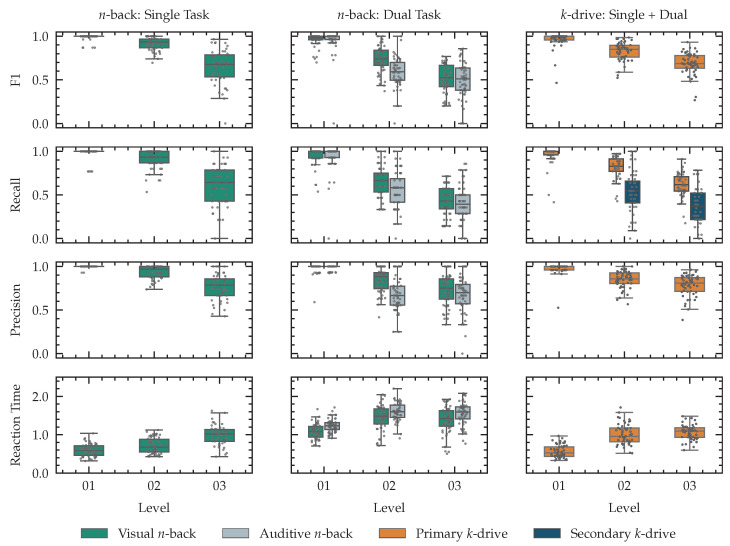
Recorded performance metrics during all conducted tests. For single-task *n*-back performance, we report F1-score, recall, precision and reaction time (**first column**). If a dual task was executed, we report these results as an additional box for the corresponding level (**second column**). For *k*-drive we report the same metrics for the single-task tests. As we only counted positive hits for our second task during dual-task *k*-drive, we report recall as a performance metric. (**last column**).

**Figure 12 sensors-23-00340-f012:**
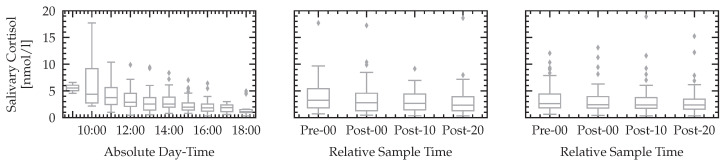
The start of each experiment was executed with a given daytime schedule described in Section 2.5. Some subjects required more time for task comprehension or were repeating the training phase multiple times. This leads to different absolute day-times of cortisol measurements while complying with the relative schedule described in Section 2.5. The first plot shows all measured cortisol values, throughout the day (accumulated one hour intervals). The second plot visualizes the cortisol values for the *n*-back experiment, while the last plot shows the results for *k*-drive.

**Figure 13 sensors-23-00340-f013:**
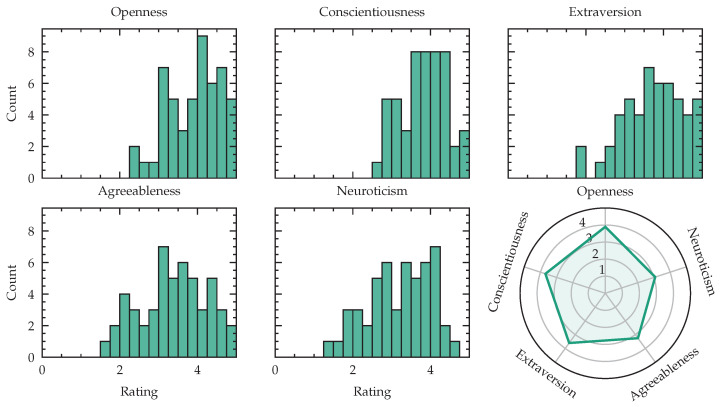
Big-Five personality traits: openness, conscientiousness, extraversion, agreeableness, neuroticism visualized as histogram and radar plot of mean values.

**Figure 14 sensors-23-00340-f014:**
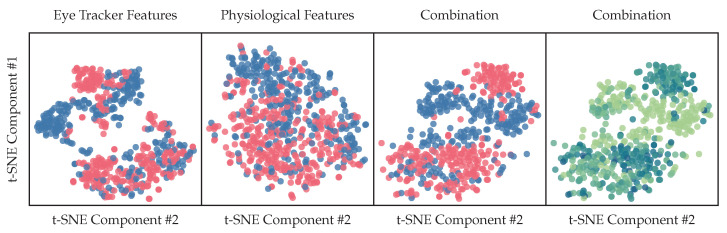
Visualization of 2-component t-SNE representations for computed biosignal- and eye-tracker-based features. The first plot shows t-SNE representation with eye-tracker features only, the second plot depicts only biosignal-based features and the third plot visualizes the combinations of both feature sets. The last plot presents the same representations from the third plot, but color coded as a linear combination of reaction time, recall and NASA-TLX score as color intensity.

**Figure 15 sensors-23-00340-f015:**
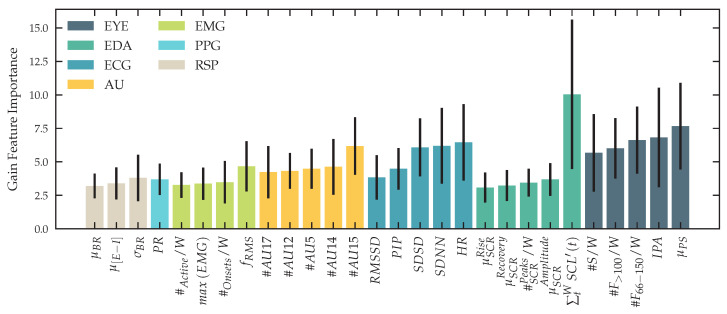
XGBoost provides among others *gain* as feature importance metric. We report the features with the five highest *gain* importance values, if available. *Gain* captures the feature importance by measuring the relative contribution of each feature in every tree to the final predictions, where higher values compared to other features indicate higher importance. Gain values are reported from models reported in Table 8 with bio-psychological features, action units and eye tracker-based features on combined (dual and single task load) with data from both experiments (10-fold-nested-cross-validation).

**Table 1 sensors-23-00340-t001:** List of extracted expert and statistical features from the recorded biosignal modalities.

Modality	Features	Reference	Description
ECG	HR ^1^	[33]	Mean heart rate
	SDSD	[33,71]	Standard deviation of successive NN intervals
	SDNN	[33,71]	Standard deviation of NN intervals
	RMSDD	[33,71]	Root mean square of successive differences
	LF, HF	[33,71]	Low and high-frequency component in the range of 0.05–0.15 Hz and 0.15–0.4 Hz of the Welch spectrum
	$LF_{n}$, $HF_{n}$	[33,71]	Normalized HF and LF spectrum
	LF/HF	[33,71]	Ratio of the LF and HF component
	SD1/SD2	[33,71]	Ratio of SD1 and SD2 as standard deviation along the identity lines of a Poincaré plot
	PSS	[78]	Percentage of short segments.
	PIP	[78]	Percentage of inflection points in NN intervals
PPG	PR ^1^	[33]	Pulse rate, closely related to ECG HR (electrical vs. mechanical)
EDA	$\sum_{t}^{W}SCL^{\prime }\left(t\right)$	[33,79,80]	Change of the Skin conductance level within a window
	${\#}_{SCR}^{Peaks}/W$	[33,73,80,81]	Number of peaks as measure of the phasic component
	$\mu_{SCR}^{Amplitude}$	[33,73]	Mean amplitude values of SCR peaks
	$\mu_{SCR}^{Rise}$	[33]	Mean rise time of SCR peaks
	$\mu_{SCR}^{Recovery}$	[33]	Mean recovery time to 50 percent of the maximal peak amplitude
EMG	$f_{RMS}\left(EMG\right)$	[82]	Root mean square of the EMG signal
	${\#}_{Onsets}\left(EMG\right)/W$	[83,84]	Number of onsets per minute
	${\#}_{Active}\left(EMG\right)/W$	[83,84]	Fraction of high activity (above threshold) per minute
	$max\left(EMG\right)$	[83]	Maximal amplitude of the EMG signal
SKT	$\mu_{T}$, $\sigma_{T}$	-	Mean and standard deviation of the skin temperature
	$min_{T}$, $max_{T}$	-	Minimal and maximal of the temperature
	$\sum T^{\prime }\left(t\right)/Time$	-	Increasing/Decreasing temperature per minute.
RSP	$\mu_{BR},\sigma_{BR}$	[77]	Mean and standard deviation of Breathing Rate (BR)
	$\mu_{\left[E-I\right]}$	[77]	Mean of the ratio of inhalation and exhalation amplitude values

^1^ We denote the mean *μ* and standard deviation *σ* for all modalities, except for ECG features, we use *HR*, *SDNN* and PPG we use *PR* since these notations are more common in the affective computing domain.

**Table 2 sensors-23-00340-t002:** List of extracted expert and statistical eye tracking features.

Concept	Features	Reference	Description
Fixations	$\#F_{>100}/W$	[85,94]	Fixation count per window for range >100 ms
	$\#F_{66–150}/W$	[90]	Fixation count per window for range 66–150 ms
	$\#F_{300–500}/W$	[90]	Fixation count per window for range 300–500 ms
	$\#F_{>1000}/W$	[90]	Fixation count per window for range >1000 ms
	$\mu_{FD}$	[85,94]	Mean fixation duration
	$med_{FD}$	[90]	Median fixation duration
Saccades	#S/W	[85]	Saccade count per window
	$\mu_{SA}$	[85]	Mean saccade amplitude
	$\mu_{SD}$	[85]	Mean saccade duration
	$med_{SD}$	-	Median saccade duration
Blinks	#B/W	[85]	Blink count per window
	$\mu_{BD}$	[94]	Mean blink duration
	$med_{BD}$	-	Median blink duration
Pupil	$\mu_{PS}$	[94]	Mean pupil size
	IPA	[93,94]	Index of pupillary activity based on wavelet transformation

**Table 3 sensors-23-00340-t003:** List of extracted action units.

Action Unit Number	Reference Literature	Facial Action Coding System Name
AU1	[28,29,102,103]	Inner Brow Raiser
AU2	[28,29,102,103]	Outer Brow Raiser
AU4	[28,29,102,103]	Brow Lowerer
AU5	[28,102,103]	Upper Lid Raiser
AU6	[28,29,102,103]	Cheek Raiser
AU7	[28,29,102,103]	Lid Tightener
AU9	[28,102,103]	Nose Wrinkler
AU10	[28,29,102,103]	Upper Lip Raiser
AU12	[28,29,102,103]	Lip Corner Puller
AU14	[28,29,102,103]	Dimpler
AU15	[28,29,102,103]	Lip Corner Depressor
AU17	[28,29,102,103]	Chin Raiser
AU20	[28,102]	Lip Stretcher
AU23	[28,29,102,103]	Lip Tightener
AU24	[102]	Lip Pressor
AU25	[28,29,102,103]	Lip Part
AU26	[28,102,103]	Jaw Drop
AU28	[29,102]	Lip Suck
AU43	[28,29,102] ^1^	Eyes Closed

^1^ Used closely related AU45 - Eyes Open vs. Eyes Closed.

**Table 4 sensors-23-00340-t004:** Statistical evaluation of biosignal and eye-tracker based features during *n*-back experiment. The first column denotes the modalities, followed by the computed feature. The features are described in Section 2.8 for all bio-psychological features and all eye-tracker features are described in Section 2.9. The omnibus test and post hoc analysis are described in Section 2.12. All reported *p*-values are adjusted to compensate for multiple comparison problem using Bonferroni-Holm correction. For features that show a significant change (*p*-value below α=0.05) in the omnibus test, we conducted a two-sided t-test between a baseline phase and the 2/3-back for the visual-only and dual-task load. If the result of the post hoc test is significant, we report if the mean feature value is increasing (↑) or decreasing (↓).

Modality	Feature	Post Hoc Adjusted *p*-Values				
		Omnibus Test	Baseline Tutorial			
			2-back Single	3-back Single	2-back Dual	3-back Dual
ECG	HR	<0.001	<0.001(↑)	<0.001(↑)	<0.181(↑)	<1.000(↑)
	SDSD	<0.001	<1.000(↑)	<0.008(↓)	<0.563(↑)	<1.000(↑)
	SDNN	<0.001	<0.010(↓)	<0.244(↑)	<0.174(↑)	<1.000(↑)
	RMSSD	<0.001	<1.000(↑)	<0.008(↓)	<0.563(↑)	<1.000(↑)
	LF	<0.001	<0.011(↓)	<0.003(↓)	<0.018(↓)	<1.000(↑)
	HF	<0.001 ^†^	<1.000(↑)	<1.000(↑)	<1.000(↑)	<1.000(↑)
	$LF_{n}$	<0.001 ^†^	<0.001(↓)	<0.161(↑)	<1.000(↑)	<1.000(↑)
	$HF_{n}$	<0.001 ^†^	<0.001(↑)	<0.161(↑)	<1.000(↑)	<1.000(↑)
	LF/HF	<0.001 ^†^	<0.001(↓)	<0.210(↑)	<0.670(↑)	<1.000(↑)
	SD1/SD2	<0.001	<1.000(↑)	<1.000(↑)	<1.000(↑)	<1.000(↑)
	PSS	<0.001	<0.531(↑)	<1.000(↑)	<1.000(↑)	<1.000(↑)
	PIP	<0.001	<0.111(↑)	<1.000(↑)	<0.057(↑)	<1.000(↑)
PPG	PR	<0.029	<1.000(↑)	<1.000(↑)	<1.000(↑)	<1.000(↑)
EDA	$\mu_{SCR}^{Amplitude}$	<0.186	-	-	-	-
	$\sum_{t}^{W}SCL^{\prime }\left(t\right)$	<0.130 ^†^	-	-	-	-
	${\#}_{SCR}^{Peaks}/W$	<0.604	-	-	-	-
	$\mu_{SCR}^{Rise}$	<0.132	-	-	-	-
	$\mu_{SCR}^{Recovery}$	<0.396 ^†^	-	-	-	-
EMG	${\#}_{Onsets}/W$	<0.271	-	-	-	-
	${\#}_{Active}/W$	<0.132	-	-	-	-
	$max\left(EMG\right)$	<0.029	<1.000(↑)	<1.000(↑)	<1.000(↑)	<1.000(↑)
	$f_{RMS}$	<0.189	-	-	-	-
SKT	$\mu_{T}$	<0.008	<0.013(↓)	<0.125(↑)	<0.144(↑)	<1.000(↑)
	$\sigma_{T}$	<0.008	<0.013(↓)	<0.125(↑)	<0.144(↑)	<1.000(↑)
	$min_{T}$	<0.017	<0.016(↓)	<0.080(↑)	<0.059(↑)	<1.000(↑)
	$max_{T}$	<0.192	-	-	-	-
	$\sum T^{\prime }\left(t\right)/T$	<0.001	<0.987(↑)	<0.563(↑)	<0.031(↑)	<0.008(↑)
RSP	$\mu_{BR}$	<0.001	<0.001(↑)	<0.129(↑)	<1.000(↑)	<1.000(↑)
	$\sigma_{BR}$	<0.060 ^†^	-	-	-	-
	$\mu_{\left[E-I\right]}$	<0.271	-	-	-	-
EYE	$\#F_{>100}/W$	<0.001	<1.000(↑)	<1.000(↑)	<1.000(↑)	<1.000(↑)
	$\#F_{66-150}/W$	<0.153 ^†^	-	-	-	-
	$\#F_{300-500}/W$	<0.001	<1.000(↑)	<1.000(↑)	<1.000(↑)	<1.000(↑)
	$\#F_{>1000}/W$	<0.001	<1.000(↑)	<1.000(↑)	<0.016(↓)	<1.000(↑)
	$\mu_{FD_{>100}}$	<0.035	<1.000(↑)	<1.000(↑)	<0.245(↑)	<0.085(↑)
	$med_{FD_{>100}}$	<0.006	<1.000(↑)	<0.676(↑)	<0.128(↑)	<1.000(↑)
	#S/W	<0.001	<1.000(↑)	<1.000(↑)	<1.000(↑)	<1.000(↑)
	$\mu_{SA}$	<0.001	<1.000(↑)	<0.032(↓)	<1.000(↑)	<1.000(↑)
	$\mu_{SD}$	<0.604	-	-	-	-
	$med_{SD}$	<0.001 ^†^	<1.000(↑)	<1.000(↑)	<1.000(↑)	<1.000(↑)
	#B/W	<0.159	-	-	-	-
	$\mu_{BD}$	<0.001	<1.000(↑)	<1.000(↑)	<1.000(↑)	<1.000(↑)
	$med_{BD}$	<0.076	-	-	-	-
	$\mu_{PS}$	<0.076	-	-	-	-
	IPA	<0.001	<0.001(↑)	<0.001(↑)	<0.001(↑)	<0.001(↑)

^†^ Non-parametric Friedman test.

**Table 5 sensors-23-00340-t005:** Tabular view of statistical results for features computed during *k*-drive, indicating significant changes, for multiple modalities. All statistical tests are conducted according to Section 2.12. *p*-values are adjusted using Bonferroni-Holm. Post Hoc tests are performed for k∈{1,2,3}-drive levels and driving tutorial baseline.

Modality	Feature	Post Hoc Adjusted *p*-Values			
		Omnibus Test	Baseline Tutorial		
			1-Drive	2-Drive	3-Drive
ECG	HR	<0.001	<0.076(↑)	<0.001(↑)	<0.001(↑)
	SDSD	<0.003	<0.008(↓)	<0.001(↓)	<0.001(↓)
	SDNN	<0.001	<0.002(↓)	<0.001(↓)	<0.001(↓)
	RMSSD	<0.003	<0.008(↓)	<0.001(↓)	<0.001(↓)
	LF	<0.074	-	-	-
	HF	<0.201	-	-	-
	$LF_{n}$	<0.010	<1.000(↑)	<1.000(↑)	<1.000(↑)
	$HF_{n}$	<0.010	<1.000(↑)	<1.000(↑)	<1.000(↑)
	LF/HF	<0.243 ^†^	-	-	-
	SD1/SD2	<0.001	<1.000(↑)	<1.000(↑)	<1.000(↑)
	PSS	<0.001	<1.000(↑)	<0.438(↑)	<1.000(↑)
	PIP	<0.001	<1.000(↑)	<1.000(↑)	<1.000(↑)
PPG	PR	<0.418	-	-	-
EDA	$\mu_{SCR}^{Amplitude}$	<0.044	<1.000(↑)	<0.806(↑)	<0.264(↑)
	$\sum_{t}^{W}SCL^{\prime }\left(t\right)$	<1.000	-	-	-
	${\#}_{SCR}^{Peaks}/W$	<0.001	<1.000(↑)	<0.012(↑)	<0.070(↑)
	$\mu_{SCR}^{Rise}$	<0.022	<1.000(↑)	<0.110(↑)	<1.000(↑)
	$\mu_{SCR}^{Recovery}$	<0.013 ^†^	<0.102(↑)	<1.000(↑)	<1.000(↑)
EMG	${\#}_{Onsets}/W$	<0.001 ^†^	<1.000(↑)	<1.000(↑)	<0.176(↑)
	${\#}_{Active}/W$	<0.074	-	-	-
	$max\left(EMG\right)$	<0.201	-	-	-
	$f_{RMS}$	<0.001 ^†^	<1.000(↑)	<0.001(↑)	<0.001(↑)
SKT	$\mu_{T}$	<0.001	<0.008(↓)	<0.001(↓)	<0.004(↓)
	$\sigma_{T}$	<0.001	<0.008(↓)	<0.001(↓)	<0.004(↓)
	$min_{T}$	<0.001	<0.004(↓)	<0.001(↓)	<0.005(↓)
	$max_{T}$	<0.001	<0.001(↓)	<0.001(↓)	<0.001(↓)
	$\sum T^{\prime }\left(t\right)/T$	<0.002	<0.001(↑)	<0.657(↑)	<0.233(↑)
RSP	$\mu_{BR}$	<0.001	<1.000(↑)	<1.000(↑)	<1.000(↑)
	$\sigma_{BR}$	<0.031 ^†^	<1.000(↑)	<1.000(↑)	<1.000(↑)
	$\mu_{\left[E-I\right]}$	<1.000	-	-	-
EYE	$\#F_{>100}/W$	<0.001	<1.000(↑)	<0.001(↓)	<0.001(↓)
	$\#F_{66-150}/W$	<0.001	<0.772(↑)	<0.001(↓)	<0.001(↓)
	$\#F_{300-500}/W$	<0.001 ^†^	<0.672(↑)	<1.000(↑)	<1.000(↑)
	$\#F_{>1000}/W$	<0.002 ^†^	<1.000(↑)	<1.000(↑)	<1.000(↑)
	$\mu_{FD_{>100}}$	<0.001	<0.051(↑)	<0.001(↑)	<0.001(↑)
	$med_{FD_{>100}}$	<0.001	<1.000(↑)	<0.001(↑)	<0.001(↑)
	#S/W	<0.001 ^†^	<1.000(↑)	<0.712(↑)	<0.244(↑)
	$\mu_{SA}$	<0.001	<0.001(↓)	<0.001(↑)	<0.001(↑)
	$\mu_{SD}$	<0.418 ^†^	-	-	-
	$med_{SD}$	<0.020 ^†^	<0.404(↑)	<1.000(↑)	<1.000(↑)
	#B/W	<0.001	<0.184(↑)	<0.089(↑)	<0.048(↓)
	$\mu_{BD}$	<0.025	<0.020(↑)	<1.000(↑)	<1.000(↑)
	$med_{BD}$	<0.006	<0.003(↑)	<1.000(↑)	<1.000(↑)
	$\mu_{PS}$	<0.001	<0.173(↑)	<0.449(↑)	<1.000(↑)
	IPA	<0.001	<1.000(↑)	<0.001(↑)	<0.001(↑)

^†^ Non-parametric Friedman test.

**Table 6 sensors-23-00340-t006:** Behavioral changes in FACS action unit for 2/3-back tests for a single and dual-task load. Every action unit occurrence is counted over every frame within a given window.

Modality	Feature	Adjusted *p*-Values				
		Omnibus Test	Baseline Tutorial			
			2-back Single	3-back Single	2-back Dual	3-back Dual
AUS	#AU1	<0.107	-	-	-	-
	#AU2	<0.001	<1.000(↑)	<1.000(↑)	<1.000(↑)	<1.000(↑)
	#AU4	<0.001	<1.000(↑)	<0.312(↑)	<1.000(↑)	<0.966(↑)
	#AU5	<0.001	<1.000(↑)	<1.000(↑)	<0.312(↑)	<0.741(↑)
	#AU6	<0.509	-	-	-	-
	#AU7	<0.722	-	-	-	-
	#AU9	<0.509	-	-	-	-
	#AU10	<0.509	-	-	-	-
	#AU11	<0.509	-	-	-	-
	#AU12	<0.987	-	-	-	-
	#AU14	<0.987	-	-	-	-
	#AU15	<0.004	<1.000(↑)	<0.015(↓)	<0.006(↓)	<0.002(↓)
	#AU17	<0.009	<0.587(↑)	<0.001(↓)	<0.025(↓)	<0.001(↓)
	#AU20	<0.294	-	-	-	-
	#AU23	<0.001	<1.000(↑)	<1.000(↑)	<1.000(↑)	<1.000(↑)
	#AU24	<0.009	<1.000(↑)	<1.000(↑)	<1.000(↑)	<1.000(↑)
	#AU25	<0.509	-	-	-	-
	#AU26	<0.091	-	-	-	-
	#AU28	<0.722	-	-	-	-
	#AU43	<0.002	<1.000(↑)	<1.000(↑)	<0.011(↑)	<0.206(↑)

**Table 7 sensors-23-00340-t007:** Changes in action units during k∈{1,2,3}-drive test. Increase (↑) and decrease (↓) indicate data distribution shift between single (1-drive) and dual-task (2/3-drive) load.

Modality	Feature	Adjusted *p*-Values			
		Omnibus Test	Baseline Tutorial		
			1-Drive	2-Drive	3-Drive
AUS	#AU1	<0.001	<1.000(↑)	<1.000(↑)	<1.000(↑)
	#AU2	<0.001	<0.606(↑)	<0.023(↑)	<0.073(↑)
	#AU4	<0.283	-	-	-
	#AU5	<0.001	<0.051(↑)	<0.001(↑)	<0.001(↑)
	#AU6	<0.268 ^†^	-	-	-
	#AU7	<0.554	-	-	-
	#AU9	<0.164	-	-	-
	#AU10	<0.078	-	-	-
	#AU11	<0.867	-	-	-
	#AU12	<0.003	<1.000(↑)	<0.003(↓)	<0.027(↓)
	#AU14	<0.043 ^†^	<1.000(↑)	<0.049(↓)	<0.073(↑)
	#AU15	<0.148	-	-	-
	#AU17	<0.003	<1.000(↑)	<1.000(↑)	<1.000(↑)
	#AU20	<0.001	<0.002(↓)	<0.075(↑)	<0.314(↑)
	#AU23	<0.011	<0.888(↑)	<0.350(↑)	<0.252(↑)
	#AU24	<0.043	<1.000(↑)	<0.008(↑)	<0.728(↑)
	#AU25	<0.867	-	-	-
	#AU26	<0.006	<1.000(↑)	<0.706(↑)	<1.000(↑)
	#AU28	<0.114	-	-	-
	#AU43	<0.268	-	-	-

^†^ Non-parametric Friedman test.

**Table 8 sensors-23-00340-t008:** Tabular data visualization of F1-Score and AUC scores for binary classification task. Rows contain various sets of features used for classification: Using only one modality (ECG, EDA, etc.) or combinations of various modalities: *Bio* containing all biosignal modalities, described in Section 2.8, *AU’s* containing all action units extracted from video data and *Eye* containing all eye tracker features. The columns are separated into experiments described in Section 2.3 and Section 2.4 and the combination of both experiments. The different groups for this two-level classification, detecting low and high cognitive load are described in Section 2.13. The second-level hierarchical columns differentiate between single and dual-task load (e.g., for the n-back task: visual vs. visual + auditive) and the combinations of similar cognitive load levels by intensities in *Comb.*, described in Section 2.13. All results were acquired using the evaluation pipeline described in Section 2.14 with nested cross-validation protocol and *XGBoost* as model for classification.

	*n*-back						*k*-Drive						Both					
	Single		Dual		Comb.		Single		Dual		Comb.		Single		Dual		Comb.	
	$F_{1}$	AUC	$F_{1}$	AUC	$F_{1}$	AUC	$F_{1}$	AUC	$F_{1}$	AUC	$F_{1}$	AUC	$F_{1}$	AUC	$F_{1}$	AUC	$F_{1}$	AUC
PPG	0.61 ± 0.10	0.66 ± 0.09	0.48 ± 0.21	0.61 ± 0.09	0.53 ± 0.10	0.60 ± 0.09	0.58 ± 0.12	0.65 ± 0.12	0.59 ± 0.07	0.61 ± 0.12	0.53 ± 0.12	0.57 ± 0.06	0.52 ± 0.16	0.62 ± 0.09	0.54 ± 0.07	0.57 ± 0.04	0.52 ± 0.09	0.55 ± 0.06
Respiration	0.61 ± 0.14	0.71 ± 0.13	0.64 ± 0.12	0.69 ± 0.10	0.62 ± 0.11	0.65 ± 0.12	0.61 ± 0.11	0.68 ± 0.11	0.65 ± 0.11	0.72 ± 0.08	0.70 ± 0.10	0.76 ± 0.10	0.64 ± 0.05	0.65 ± 0.04	0.62 ± 0.09	0.67 ± 0.07	0.63 ± 0.05	0.65 ± 0.06
EMG	0.62 ± 0.09	0.66 ± 0.08	0.61 ± 0.09	0.60 ± 0.10	0.62 ± 0.08	0.62 ± 0.05	0.65 ± 0.11	0.69 ± 0.11	0.69 ± 0.09	0.75 ± 0.09	0.69 ± 0.08	0.73 ± 0.05	0.60 ± 0.18	0.66 ± 0.05	0.62 ± 0.10	0.66 ± 0.05	0.61 ± 0.10	0.67 ± 0.06
ECG	0.67 ± 0.11	0.73 ± 0.11	0.67 ± 0.08	0.73 ± 0.06	0.64 ± 0.14	0.71 ± 0.10	0.71 ± 0.15	0.87 ± 0.14	0.73 ± 0.07	0.88 ± 0.10	0.79 ± 0.09	0.87 ± 0.07	0.67 ± 0.11	0.74 ± 0.09	0.70 ± 0.08	0.78 ± 0.07	0.70 ± 0.07	0.75 ± 0.06
EDA	0.63 ± 0.07	0.69 ± 0.10	0.69 ± 0.06	0.76 ± 0.08	0.67 ± 0.10	0.71 ± 0.08	0.93 ± 0.11	0.96 ± 0.06	0.87 ± 0.12	0.97 ± 0.04	0.79 ± 0.04	0.88 ± 0.03	0.72 ± 0.07	0.75 ± 0.06	0.73 ± 0.08	0.81 ± 0.05	0.63 ± 0.05	0.68 ± 0.04
Biosignals	0.70 ± 0.13	0.79 ± 0.11	0.70 ± 0.09	0.78 ± 0.08	0.69 ± 0.07	0.77 ± 0.06	0.88 ± 0.08	0.97 ± 0.07	0.92 ± 0.04	1.00 ± 0.01	0.86 ± 0.06	0.96 ± 0.03	0.73 ± 0.07	0.81 ± 0.07	0.76 ± 0.08	0.87 ± 0.06	0.74 ± 0.05	0.81 ± 0.05
Eye Tracker	0.79 ± 0.08	0.92 ± 0.05	0.74 ± 0.08	0.82 ± 0.10	0.76 ± 0.06	0.86 ± 0.04	0.68 ± 0.12	0.82 ± 0.13	0.88 ± 0.07	0.96 ± 0.04	0.79 ± 0.05	0.89 ± 0.06	0.73 ± 0.11	0.86 ± 0.07	0.78 ± 0.06	0.90 ± 0.04	0.73 ± 0.07	0.84 ± 0.05
Action Units	0.60 ± 0.13	0.68 ± 0.06	0.70 ± 0.06	0.72 ± 0.10	0.54 ± 0.14	0.70 ± 0.06	0.64 ± 0.15	0.74 ± 0.09	0.68 ± 0.12	0.78 ± 0.11	0.71 ± 0.06	0.77 ± 0.08	0.52 ± 0.12	0.65 ± 0.09	0.59 ± 0.16	0.74 ± 0.08	0.64 ± 0.10	0.69 ± 0.07
Bio, AU’s	0.70 ± 0.14	0.80 ± 0.11	0.71 ± 0.11	0.85 ± 0.06	0.70 ± 0.08	0.80 ± 0.06	0.89 ± 0.07	0.97 ± 0.05	0.91 ± 0.05	0.99 ± 0.03	0.85 ± 0.07	0.96 ± 0.03	0.73 ± 0.05	0.81 ± 0.05	0.80 ± 0.08	0.90 ± 0.06	0.73 ± 0.08	0.84 ± 0.06
Eye, AU’s	0.80 ± 0.09	0.93 ± 0.06	0.78 ± 0.06	0.87 ± 0.07	0.77 ± 0.06	0.87 ± 0.05	0.68 ± 0.20	0.88 ± 0.11	0.90 ± 0.06	0.98 ± 0.03	0.80 ± 0.09	0.91 ± 0.07	0.75 ± 0.08	0.87 ± 0.05	0.81 ± 0.07	0.91 ± 0.04	0.77 ± 0.05	0.86 ± 0.05
Bio, Eye	0.82 ± 0.07	0.93 ± 0.06	0.75 ± 0.12	0.86 ± 0.09	0.78 ± 0.06	0.87 ± 0.05	0.91 ± 0.06	0.97 ± 0.05	0.92 ± 0.04	0.99 ± 0.02	0.87 ± 0.09	0.95 ± 0.04	0.81 ± 0.08	0.92 ± 0.06	0.83 ± 0.08	0.92 ± 0.06	0.81 ± 0.06	0.89 ± 0.04
Bio, Eye, AU’s	0.82 ± 0.09	0.93 ± 0.06	0.78 ± 0.09	0.88 ± 0.09	0.77 ± 0.07	0.88 ± 0.05	0.88 ± 0.09	0.96 ± 0.06	0.94 ± 0.05	0.99 ± 0.01	0.89 ± 0.06	0.96 ± 0.03	0.84 ± 0.06	0.92 ± 0.06	0.83 ± 0.05	0.92 ± 0.04	0.82 ± 0.06	0.90 ± 0.04

**Table 9 sensors-23-00340-t009:** Results of three-level-intensity classification tasks, treating different levels as separate classes (columns). For three different combinations of features, based on biosignals alone, only eye-tracker data and the combination of features based on multiple modalities: biosignals, eye-tracker-based action units (rows). We report the confusion matrix for the conducted experiments, using only *n*-back levels, only Drive levels and a combination of both experiments. The grouping of different phases for all experiments is described in Section 2.13. The rows correspond to the true class.

		*n*-back			*k*-Drive			Both		
		Low	Med.	High	Low	Med.	High	Low	Med.	High
Biosignals	Low	0.57 ± 0.23	0.17 ± 0.12	0.26 ± 0.14	0.82 ± 0.07	0.08 ± 0.07	0.09 ± 0.06	0.55 ± 0.10	0.21 ± 0.05	0.25 ± 0.08
	Medium	0.29 ± 0.15	0.49 ± 0.12	0.22 ± 0.18	0.09 ± 0.16	0.51 ± 0.19	0.40 ± 0.20	0.24 ± 0.13	0.42 ± 0.08	0.34 ± 0.07
	High	0.21 ± 0.10	0.24 ± 0.05	0.55 ± 0.09	0.07 ± 0.11	0.51 ± 0.26	0.42 ± 0.23	0.22 ± 0.08	0.27 ± 0.06	0.51 ± 0.07
Eye Tracker	Low	0.86 ± 0.12	0.07 ± 0.11	0.06 ± 0.08	0.84 ± 0.11	0.07 ± 0.06	0.09 ± 0.06	0.77 ± 0.16	0.09 ± 0.08	0.13 ± 0.09
	Medium	0.17 ± 0.22	0.49 ± 0.12	0.33 ± 0.19	0.00 ± 0.00	0.70 ± 0.19	0.30 ± 0.19	0.15 ± 0.16	0.55 ± 0.17	0.31 ± 0.14
	High	0.09 ± 0.07	0.32 ± 0.05	0.58 ± 0.05	0.02 ± 0.06	0.27 ± 0.22	0.71 ± 0.22	0.07 ± 0.07	0.33 ± 0.04	0.60 ± 0.08
Bio, Eye, AU’s	Low	0.77 ± 0.17	0.10 ± 0.09	0.13 ± 0.11	0.92 ± 0.07	0.02 ± 0.04	0.06 ± 0.05	0.81 ± 0.12	0.08 ± 0.07	0.11 ± 0.06
	Medium	0.13 ± 0.12	0.67 ± 0.22	0.21 ± 0.17	0.03 ± 0.05	0.71 ± 0.18	0.26 ± 0.14	0.18 ± 0.15	0.50 ± 0.09	0.32 ± 0.10
	High	0.08 ± 0.05	0.27 ± 0.06	0.65 ± 0.07	0.03 ± 0.06	0.31 ± 0.17	0.66 ± 0.19	0.07 ± 0.06	0.32 ± 0.05	0.61 ± 0.05

**Table 10 sensors-23-00340-t010:** Comparison of different classifiers for all binary tasks. All classifiers have been trained using the complete set of features: bio-signals, eye-tracker data and action units from videos. The results in this table correspond to the last row in Table 8 for *XGBoost* based classification. *kNN* is a k-Nearest-Neighbor classifier, *SVM (rbf)* a support vector machine with a radial kernel and *SVM (lin)* is a linear support vector machine. All results have been acquired using nested-cross-validation.

	*n*-back						*k*-Drive						Both					
	Single		Dual		Comb.		Single		Dual		Comb.		Single		Dual		Comb.	
	$F_{1}$	AUC	$F_{1}$	AUC	$F_{1}$	AUC	$F_{1}$	AUC	$F_{1}$	AUC	$F_{1}$	AUC	$F_{1}$	AUC	$F_{1}$	AUC	$F_{1}$	AUC
kNN	0.71 ± 0.09	0.78 ± 0.11	0.69 ± 0.11	0.72 ± 0.09	0.71 ± 0.09	0.76 ± 0.06	0.71 ± 0.10	0.86 ± 0.10	0.83 ± 0.07	0.94 ± 0.06	0.80 ± 0.05	0.90 ± 0.04	0.72 ± 0.05	0.79 ± 0.08	0.76 ± 0.07	0.84 ± 0.06	0.75 ± 0.03	0.81 ± 0.03
XGB	0.82 ± 0.09	0.93 ± 0.06	0.78 ± 0.09	0.88 ± 0.09	0.77 ± 0.07	0.88 ± 0.05	0.88 ± 0.09	0.96 ± 0.06	0.94 ± 0.05	0.99 ± 0.01	0.89 ± 0.06	0.96 ± 0.03	0.84 ± 0.06	0.92 ± 0.06	0.83 ± 0.05	0.92 ± 0.04	0.82 ± 0.06	0.90 ± 0.04
SVM (rbf)	0.79 ± 0.11	0.85 ± 0.10	0.69 ± 0.13	0.79 ± 0.06	0.73 ± 0.07	0.83 ± 0.07	0.80 ± 0.14	0.89 ± 0.09	0.88 ± 0.10	0.96 ± 0.06	0.86 ± 0.08	0.92 ± 0.07	0.72 ± 0.10	0.81 ± 0.10	0.80 ± 0.06	0.88 ± 0.04	0.77 ± 0.05	0.84 ± 0.05
SVM (lin)	0.75 ± 0.13	0.76 ± 0.25	0.69 ± 0.16	0.75 ± 0.22	0.73 ± 0.08	0.82 ± 0.09	0.63 ± 0.35	0.72 ± 0.34	0.89 ± 0.06	0.97 ± 0.05	0.84 ± 0.04	0.92 ± 0.05	0.73 ± 0.09	0.81 ± 0.09	0.80 ± 0.06	0.87 ± 0.06	0.76 ± 0.05	0.84 ± 0.07

**Table 11 sensors-23-00340-t011:** MSE and $R^{2}$ score of regression tasks using *performance* measures, *subjective* feedback and the linear combination of both as the target value for *n*-back, *k*-drive and the combination of both experiments.

	*n*-back		*k*-Drive		Both	
	$R^{2}$	MSE	$R^{2}$	MSE	$R^{2}$	MSE
Performance	0.50 ± 0.11	0.038 ± 0.008	0.53 ± 0.12	0.030 ± 0.009	0.51 ± 0.07	0.037 ± 0.005
Subjective	0.43 ± 0.13	0.043 ± 0.013	0.59 ± 0.06	0.030 ± 0.006	0.48 ± 0.08	0.041 ± 0.011
Combination	0.53 ± 0.10	0.041 ± 0.010	0.58 ± 0.08	0.033 ± 0.005	0.54 ± 0.08	0.041 ± 0.009

## Data Availability

Interested parties may use a subset (30 subjects) of the data presented here, after returning a signed End User License Agreement (EULA), to the Fraunhofer Institute of Integrated Circuits. The signed EULA should be returned in digital format by sending it to *adabase@iis.fraunhofer.de*. The usage of the dataset for any nonacademic purpose is prohibited. Nonacademic purposes include, but are not limited to: proving the efficiency of commercial systems, training or testing of commercial systems, selling data from the dataset, creating military applications and developing governmental systems used in public spaces.

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
