# Peer review of "ADABase: A Multimodal Dataset for Cognitive Load Estimation"

_sensors, 2022, doi:10.3390/s23010340_

Round 1

Reviewer 1 Report

This is a well-written paper which will be of much use to the MMLA community. I appreciate the authors' attention to detail and exposition. Below are some queries and suggestions for the authors to consider. 

1) How were missing data handled for each modality and what were the percentage of such data for each modality?

2) Why are the authors only releasing data for 30 people. This is a serious limitation as the utility of the dataset becomes severely limited with such small N. I would request the authors to release the complete dataset for transparency and usability. 

3) Fixation durations in eye tracking data are always right skewed. Although people have traditionally used mean as a central tendency measure for such data the limitation of the mean is well documented. Negi and Mitra 2020 further pointed out that the underlying cause of fixation durations can be quite different and hence should be treated differently. Hence, I urge the authors to provide median and count of fixations for the ranges suggested in Negi and Mitra 2020 as well, if possible. At the very least count of fixations greater than 1000 ms should be provided.

4) Please pay careful attention to citation style (E.g., in line 80 the sentence stops abruptly with bracketed citation and in line 101 there is no year with the citation). Minor language issues like calling sections chapters (line 403).

Author Response

Thank you for your feedback. We hope that these responses address your concerns and that our revised paper will be of even greater use to the research community.

1) In response to your first query, we have now clarified the handling of missing/corrupt data by providing a step-by-step description of our preprocessing, artifact rejection, and outlier removal technique. In addition to this, we have reported the percentages of available instances of every computed feature for reference as an additional section in the appendix of our publication. Our work is not focused on finding a predictor, that can handle records with missing features. The occurrence of missing modalities is statistically independent and we favor an unbiased result and therefore do not employ any imputation techniques.

2) Regarding the limited release of data: We release data for 30 subjects for multiple reasons. (a) We are planning to host a machine learning competition, utilizing the remaining data as a hidden test set, in the future. (b) We want to keep a hidden test, to enable future verification and validation of third-party results. Nonetheless, we are more than confident, that our 30-subject public split is a very valuable addition to the research community and will to this point not release a bigger portion.

3) We appreciate your suggestion to provide additional measures/features, such as median and/or three non-overlapping temporal ranges for the computation of features based on fixation duration. We remark, that our publication's main focus is not the application of features with the highest predictive power but agree with your suggestion to add median features for duration-based eye-tracking measures and features with ranges between 66-150ms, 300-500ms, and >1000ms based on Negi and Mitra 2020. Thank you for your suggestion to include those measures, we think it's a very valuable addition.

4) Thank you for pointing out the citation style and language issues in the manuscript. We have noticed the bold vs non-bold publication years format for different publication types and adopted these accordingly. We will make sure to carefully proofread the revised manuscript and address any issues before resubmitting it.

Again, thank you for your helpful feedback. We did our best to address all of your comments and queries in the revised manuscript.

Reviewer 2 Report

This study proposed a multimodal dataset for cognitive load detection. The paper is well-written and their findings would be worth for the development of driver’s cognitive load detection systems in future research. The study of the manuscript is meaningful, and the logic of the manuscript is clear. However, there are still some points that need to be clearly discussed.

It is suggested that the authors could provide an appropriate explanation for the results in the Discussion section or add a comparative analysis with related studies.

Author Response

Thank you for your review and for highlighting the strengths of our study. We are glad that you found the paper well-written and that the findings are worth considering for the development of driver's cognitive load detection systems in future research. We appreciate your suggestion to provide an appropriate explanation for the results in the Discussion section and to add a comparative analysis of findings with related studies. We added both in our revised version.
Thank you again for your review and for helping us to improve the quality of our work.